# A practical method for assigning uncertainty and improving the accuracy of alpha-ejection corrections and eU concentrations in apatite (U-Th)/He chronology

Spencer D. Zeigler[1], James R. Metcalf[1], Rebecca M. Flowers[1]

[1]Department of Geological Sciences, University of Colorado Boulder, Boulder, CO, 80309, USA

*Correspondence to*: Spencer D. Zeigler (spencer.zeigler@colorado.edu)

**Abstract.** Apatite (U-Th)/He (AHe) dating generally assumes that grains can be accurately and precisely modeled as geometrically perfect hexagonal prisms or ellipsoids in order to compute the apatite volume (V), alpha-ejection corrections ($F_T$), equivalent spherical radius ($R_{FT}$), effective uranium concentration (eU), and corrected (U-Th)/He date. It is well-known that this assumption is not true. In this work, we present a set of corrections and uncertainties for V, $F_T$, and $R_{FT}$ aimed 1) at "undoing" the systematic deviation from the idealized geometry, and 2) at quantifying the contribution of geometric uncertainty to the total uncertainty budget on eU and AHe dates. These corrections and uncertainties can be easily integrated into existing laboratory workflows at no added cost, can be routinely applied to all dated apatite, and can even be retroactively applied to published data. To quantify the degree to which real apatite deviate from geometric models, we selected 264 grains that span the full spectrum of commonly analyzed morphologies, measured their dimensions using standard 2D microscopy methods, and then acquired 3D scans of the same grains using high-resolution computed tomography (CT). We then compared our apatite 2D length, maximum width, and minimum width measurements with those determined by CT, as well as the V, $F_T$, and $R_{FT}$ values calculated from 2D-microscopy measurements with those from the 'real' 3D measurements. While our 2D length and maximum width measurements match the 3D values well, the 2D minimum width values systematically underestimate the 3D values and have high scatter. We therefore use only the 2D length and maximum width measurements to compute V, $F_T$, and $R_{FT}$. With this approach, apatite V, $F_T$, and $R_{FT}$ values are all consistently overestimated by the 2D microscopy method, requiring correction factors of 0.74-0.83 (or 17-26%), 0.91-0.99 (or 1-9%), and 0.85-0.93 (or 7-15%), respectively. The 1s uncertainties on V, $F_T$, and $R_{FT}$ are 20-23%, 1-6%, and 6-10%, respectively. The primary control on the magnitude of the corrections and uncertainties is grain geometry, with grain size exerting additional control on $F_T$ uncertainty. Application of these corrections and uncertainties to a real dataset (N = 24 AHe analyses) yields 1s analytical and geometric uncertainties of 15-16% on eU and 3-7% on the corrected date. These geometric corrections and uncertainties are substantial and should not be ignored when reporting, plotting, and interpreting AHe datasets. The Geometric Correction Method (GCM) presented here provides a simple and practical tool for deriving more accurate $F_T$ and eU values, and for incorporating this oft neglected geometric uncertainty into AHe dates.

# 1 Introduction

Apatite (U-Th)/He (AHe) dating is a widely-applied thermochronologic technique used to decipher low-temperature thermal histories. In addition to analysis of parent and daughter isotopes, the conventional whole crystal (U-Th)/He method typically includes microscopy measurements of the analyzed grain. These measurements are combined with an assumed idealized grain morphology to estimate the grain volume (V) and surface area, which in turn are used to calculate three important parameters: the alpha-ejection correction ($F_T$ value), the effective uranium concentration (eU), and the equivalent spherical radius. $F_T$ values are required for accurate dates on crystals that are not fragments, because $^4$He atoms travel ~20 µm during $\alpha$-decay and a correction is required to account for He lost by this effect (e.g., Farley et al., 1996; Ketcham et al., 2011). eU is important for accurate (U-Th)/He data interpretation because radiation damage scales with eU, which affects He retentivity (e.g., Shuster et al., 2006; Flowers et al., 2007). The equivalent spherical radius is used to approximate the diffusion domain of whole crystals, and is a standard parameter needed for diffusion modeling (here we use a sphere with an equivalent $F_T$ correction as the analyzed grain and refer to this parameter as $R_{FT}$).

It is well-recognized that there is both uncertainty and potentially systematic error associated with the microscopy approach to calculating geometric data and the parameters derived from them (Ehlers and Farley, 2003; Herman et al., 2007; Evans et al., 2008; Glotzbach et al., 2019; Cooperdock et al., 2019; Flowers et al., 2022a). Throughout this paper we use "uncertainty" to refer to the reproducibility of measurements, and "error" to refer to a systematic deviation between a measured value and the true value (JCGM, 2012). Figure 1 shows how the commonly assigned hexagonal and ellipsoidal grain geometries for apatite do not perfectly capture the true volumes and surface areas of real grains. Early work suggested that these deviations could cause as much as $\pm 25\%$ uncertainty on the $F_T$ values for hexagonal, prismatic apatite grains of 50 µm width, decreasing to <2% for grains with cross-sections of >125 µm (Ehlers and Farley, 2003). Geometric uncertainties and systematic error have also been explored using x-ray micro- or nano-computed tomography (CT), a non-destructive method that creates 3D models of scanned objects (Herman et al., 2007; Evans et al., 2008; Glotzbach et al., 2019; Cooperdock et al., 2019). These studies presented new, more comprehensive techniques for 2D apatite grain measurements (the 3D-He method of Glotzbach et al., 2019) and proposed a method to routinely acquire CT data for all dated apatite grains (Cooperdock et al., 2019).

Rigorous quantification of uncertainties and corrections for systematic error on the geometric parameters are required to represent and interpret AHe data accurately. For example, appropriate uncertainties on single-grain dates are important for deciding if data are normally distributed and thus reasonable to represent and model as a mean sample date, or if the data are "overdispersed" (e.g., Flowers et al., 2022b). Similarly, appropriate uncertainties on other parameters such as eU are needed to properly decipher AHe date vs. eU patterns. However, despite the past work addressing geometric uncertainties (e.g., Cooperdock et al., 2019; Glotzbach et al., 2019), the uncertainties on the grain's geometric information are not typically propagated into the uncertainties of the derived parameters (e.g., eU concentration, corrected (U-Th)/He date). Nor are data systematically corrected for potential error associated with grain measurements. This is largely because uncertainty and error in the geometric

parameters depend in large part on how much the real grain geometry deviates from that assumed, which may vary from grain to grain, depending on grain morphology, as well as possibly on grain size and other parameters. Moreover, although both the 3D-He method (Glotzbach et al., 2019) and the routine CT analysis approach (Cooperdock et al., 2019) would improve the accuracy and precision of geometric parameters, both add more time to the (U-Th)/He dating process, and in the case of the latter, requires regular access to CT instrumentation.

To address this problem, we present a time-efficient and straightforward "geometric correction" method to routinely correct for systematic error and to assign uncertainties to $F_T$, eU, and $R_{FT}$ values for the full spectrum of regularly analyzed apatite grain sizes and morphologies. This approach requires no additional work or cost beyond what is already done as part of most existing (U-Th)/He dating workflows. Nor does it necessitate additional microscopy measurements or routine CT analysis of grains, so it is easily adoptable by any lab or data user. Additionally, this method can be applied retroactively to previously collected data, even after the grains themselves have been dissolved and are no longer available for additional work. We first developed a simple classification system for apatite grains of varying shape and surface roughness. For 237 apatite crystals characterized by a wide range of morphology, size, age, and lithologic source, we then compared V, $F_T$, and $R_{FT}$ estimates calculated from 2D microscopy measurements with those determined by CT scans of the same grains at 0.64 μm resolution. We use these data to derive corrections for systematic error and to determine uncertainty values that can be applied to 2D V, $F_T$, and $R_{FT}$ values depending on the geometry and size of the analyzed apatite. These outcomes allow analysts to 1) correct geometric parameter values for systematic error, 2) propagate the $F_T$ uncertainty into the reported uncertainty on corrected (U-Th)/He dates, 3) propagate the V uncertainty into the reported uncertainty on eU values, and 4) report $R_{FT}$ value uncertainties that have potential to be included in thermal history modeling. We conclude by illustrating this approach with real (U-Th)/He data and discuss the implications for the accuracy and precision of (U-Th)/He datasets more broadly.

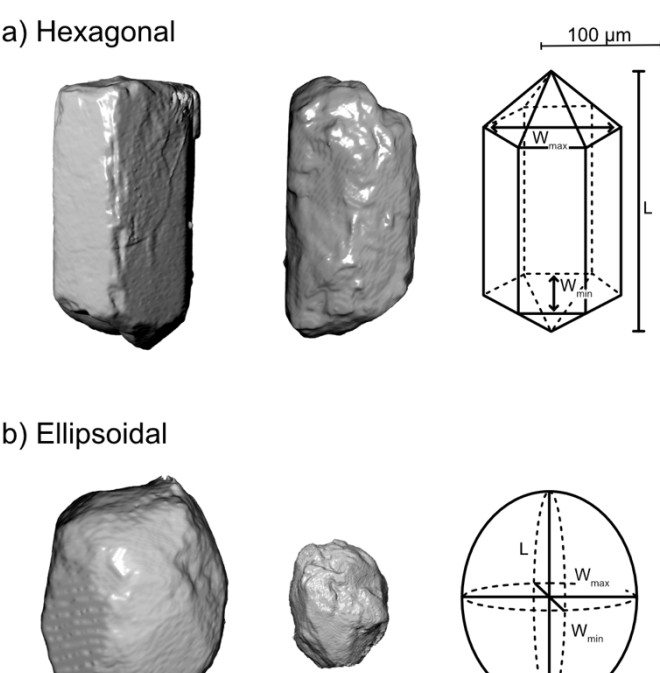

**Figure 1. 3D renderings from CT data of real apatite crystals classified as (a) hexagonal and (b) ellipsoidal versus the idealized geometry from Ketcham et al. (2011) that is used to calculate V, $F_T$, and $R_{FT}$. Scale bar is applicable to all four examples of real crystals. Note that the actual grains have geometries that are not perfectly represented by the idealized geometry. The grain length (L), maximum width ($W_{max}$), and minimum width ($W_{min}$) denoted on the schematics of the idealized geometries represent the three grain measurements made using standard 2D microscopy measurements in this study.**

## 2 Background

### 2.1 $F_T$, eU, and $R_{FT}$ values in (U-Th)/He thermochronology

An important consideration for the (U-Th)/He system is alpha ejection. During radioactive decay of the parent isotopes ($^{238}U$, $^{235}U$, $^{232}Th$, $^{147}Sm$), $^{4}He$ atoms are ejected from the parent atom (e.g., Farley et al., 1996). Alpha particles, or helium atoms, will travel a certain distance related to the density of the mineral through which they travel and the ejection energy from the parent atom. For apatite, the average stopping distances for $^{238}U$, $^{235}U$, $^{232}Th$, and $^{147}Sm$ are 18.81 µm, 21.80 µm, 22.25 µm, and 5.93 µm respectively (e.g., Ketcham et al., 2011). If the parent atom is positioned within the ejection range of the grain edge, then the He atom has a non-zero chance of being ejected from the crystal entirely. The probability of retention increases with increasing distance of the parent from the grain edge. Overall, the smaller the grain, the higher the surface area to volume ratio of the grain, and the greater percentage of He that is lost via the ejection process.

To obtain an intuitively more meaningful date, (U-Th)/He dates on crystals that retain their original grain edge are typically corrected for the He lost by alpha ejection to obtain a "corrected (U-Th)/He date". This alpha-ejection correction (or $F_T$ value) is the fraction of He that is retained in the crystal, such that an $F_T$ value of 0.70 means that an estimated 30% of He was lost from the crystal by ejection. $F_T$ is typically calculated based on the stopping distances of He in each mineral for each parent isotope, the proportion of the parent isotopes, the crystal dimensions, and an assumed idealized crystal geometry that enables one to use the crystal measurements to estimate the surface area and volume of the crystal (Farley et al., 1996). $F_T$ corrections typically assume a uniform distribution of parent isotopes; parent isotope zonation in crystals can introduce additional uncertainty into the $F_T$ correction (Farley et al., 1996; Meesters and Dunai, 2002; Hourigan et al., 2005). Additional uncertainty can also arise for broken or abraded crystals, where the magnitude of the appropriate correction can be unclear (Rahl et al., 2003; Brown et al., 2013; He and Reiners, 2022).

eU is important for (U-Th)/He thermochronology because it can be used as a proxy for radiation damage, which can have a large effect on the mineral He retentivity (e.g., Shuster et al., 2006; Flowers et al., 2007). Radiation damage can cause positive correlations between AHe date and eU for thermal histories characterized by slow cooling, partial resetting, or long residence in the helium partial retention zone. Accurate eU values depend on accurate grain volumes, because volumes are used to calculate grain masses, which in turn are used to compute parent isotope concentrations and eU (e.g., Flowers et al., 2022a).

The equivalent spherical radius is relevant for (U-Th)/He thermochronology because helium retention can depend on grain size, which must therefore be included in the diffusion modeling used to decipher thermal histories from (U-Th)/He data. The equivalent spherical radius parameter can be reported either as a sphere with the same surface area to volume ratio as the analyzed grain ($R_{SV}$), or as a sphere with the same $F_T$ value as the analyzed grain ($R_{FT}$, Ketcham et al., 2011; Cooperdock et al., 2019). Use of $R_{FT}$ is preferred, because during thermal history modeling this value yields outcomes more similar to those using the real 3D grain geometries (Ketcham et al., 2011).

## 2.2 Use of CT for $F_T$, eU, and $R_{FT}$ value determinations

Computed tomography (CT) is a high-resolution (sub-micrometer), non-destructive, 3D imaging technique based on the attenuation of x-rays through a sample. 2D cross sections ('slices') of the sample are created as x-rays pass through the sample and are then processed into 3D models. These models can be analyzed with software like Dragonfly (Object Research Systems, v.2020.2) and Blob3D (Ketcham, 2005) to extract high quality 3D geometric data like volume and surface area.

CT has been applied to improve the accuracy of geometric parameters in (U-Th)/He chronology in four studies (Herman et al., 2007; Evans et al., 2008; Glotzbach et al., 2019; Cooperdock et al., 2019). Initial work used CT data at a 6.3 µm resolution to derive $F_T$ values for 11 irregularly shaped detrital apatite grains (Herman et al., 2007). This study then dated the crystals by (U-Th)/He and combined the 3D CT models of the dated grains with an inversion algorithm to constrain a range of thermal histories.

The subsequent studies have directly compared geometric parameters determined from 2D microscopy data with 3D CT measurements of the same grains. Evans et al. (2008) calculated "effective $F_T$" values for 9 euhedral to subhedral, detrital and volcanic apatite and zircon grains using CT scans at 3.8 μm resolution and eroding the outer 20 μm of the scanned grain in 3D. Glotzbach et al. (2019) developed an improved microscopy method, called the 3D-He approach, to estimate $F_T$ values using dimensions measured from a suite of photomicrographs to simulate a 3D grain model. They acquired CT data at 1.2 μm resolution for 24 apatite grains, including rounded, pitted, broken, anhedral, subhedral, and euhedral crystals. Cooperdock et al. (2019) presented a method for regular CT characterization of grains at 4-5 μm resolution and acquired CT data for a suite of 109 high quality euhedral apatite crystals from two plutonic samples. These three studies found that the 2D data variably over- or under-estimated the 3D data for V, $F_T$, and $R_{FT}$, and estimated a range of scatter for the different parameters. These previous results are discussed in greater detail in Sect. 6.2 where we compare the outcomes of our study with this past work.

## 3 Selecting and Characterizing a Representative Apatite Suite

### 3.1 Strategy

We designed our study to ensure that we captured the range of representative apatite crystals commonly dated by the (U-Th)/He method. Our goal was to include the full spectrum of grain qualities in realistic proportions so that the study outcomes are relevant for the complete range of routinely analyzed grains rather than being biased to apatite morphologies specific to a single sample type. As described in more detail below, grain selection focused primarily on including crystals from samples encompassing a spectrum of lithology and age (Sect. 3.2), with a range of sizes (Sect. 3.3), and with variable morphology (Sect. 3.4). We ultimately selected 400 apatite grains for analysis, from which we obtained high-quality CT data for 264 crystals.

### 3.2 Selecting a Representative Sample Suite

Apatite grains were selected from eight samples that include six igneous and metamorphic rocks and two clastic sedimentary rocks with ages from Oligocene to Archean (Table 1). All samples were separated using standard crushing, density, and magnetic separation techniques. Most samples were dated previously by AHe in the CU TraIL (Thermochronology Research and Instrumentation Lab). The Oligocene Fish Canyon Tuff (sample FCT) from the San Juan Mountains in Colorado, USA is commonly used as a (U-Th)/He reference standard, with AHe dates younger than emplacement (e.g., Gleadow et al., 2015). The Eocene granitic Ipapah pluton is from the Deep Creek Range (sample DCA) of east-central Nevada, USA and yields Miocene AHe dates (unpublished data). The Cretaceous Whitehorn granodiorite (sample BF16-1) is from the Arkansas Hills in Colorado, USA and has Eocene AHe dates (Abbott et al., 2022). The Cambrian McClure Mountain syenite (sample MM1) from the Wet Mountains of south-central Colorado yields Mesozoic AHe dates (Weisberg et al., 2018). A Proterozoic granitic dike from the Baileyville drill core (sample Bail933) in northeastern Kansas, USA is

characterized by Paleozoic AHe results (Flowers and Kelley, 2011). An Archean gneiss from the Superior craton in Canada (sample C50) yields Cambrian AHe dates (TraIL unpublished data). The two detrital samples (samples 16MFS-05 and 15MFS-07) have Cretaceous depositional ages, are from the Kaikoura Range on the South Island of New Zealand, and have late Miocene to Pliocene AHe dates (Collett et al., 2019; Harbert, 2019).

**Table 1. Apatite sample information.**

| Sample Name | Unit and Lithology | Sample Age | Locality | Longitude (°W) | Latitude (°N) | GEM Categories | N[a] | Additional Geochronologic and Thermochronologic Data |
|---|---|---|---|---|---|---|---|---|
| FCT | Fish Canyon Tuff dacite | Oligocene | San Juan Mountains, Colorado, USA | -106.93 | 37.76 | A1, A2, B1 | 30 | Zircon U-Pb 28.172 ± 0.028 Ma (2s) (Schmitz and Bowring, 2001); AHe 20.8 ± 0.4 Ma (1s) (Gleadow et al., 2015) |
| DCA | Ipapah monzogranite | Eocene | Deep Creek Range East-Central Nevada, USA | -113.92 | 39.83 | A1, A2, B1, B2, C2 | 30 | Zircon U-Pb 39 Ma ± 1 Ma (Rodgers, 1989); AHe 14.3-9.6 Ma* (TRaIL unpublished data) |
| BF16-1 | Whitehorn granodiorite | Cretaceous | Arkansas Hills, Colorado, USA | -105.90 | 38.50 | A1, A2, B1, B2 | 25 | Zircon U-Pb 67.31 Ma + 0.57/-0.78 Ma (2s) (Abbey et al., 2017); AHe 47.4 ± 4.2 Ma (1s) (Abbott et al., 2022) |
| MM | McClure Mountain syenite | Cambrian | Wet Mountains, South-Central Colorado, USA | -105.47 | 38.35 | A1, A2, B1, B2 | 36 | Hornblende $^{40}Ar/^{39}Ar$ 523.2 ± 0.9 Ma (1s) (Spell and McDougall, 2003); AHe 150-70 Ma* (Weisberg et al., 2018) |
| Bail | Baileyville drill core granitic dike | Proterozoic | Northeast Kansas, USA | -96.20 | 39.90 | A1, A2, B1, B2 | 22 | Zircon U-Pb ca. 1400 Ma (Van Schmus et al., 1987); AHe 150-70 Ma* (Flowers and Kelley, 2011) |
| C50 | Superior Craton tonalitic gneiss | Archean | Superior Craton, Canada | -92.99 | 51.76 | A1, A2, B1, B2 | 47 | Zircon U-Pb 2720-2680 Ma (Hoffman, 1988); AHe 559 to 461 Ma* (TRaIL unpublished data) |
| 16MFS-05 | Marlborough Fault System sandstone | Cretaceous | Kaikōura Ranges, South Island, New Zealand | 173.69 | −42.29 | A1, A2, B1, B2, C1, C2 | 45 | Deposition 100 ± 20 Ma (1s) (Rattenbury et al., 2006), AHe 4.2 ± 0.6 Ma (1s) (Collett et al., 2019) |
| 15MFS-07 | Marlborough Fault System greywacke | Cretaceous | Kaikōura Ranges, South Island, New Zealand | 173.22 | -41.78 | B1, B2, C1, C2 | 29 | Deposition 120 ± 22 (1s) (Harbert, 2019; Mean AHe 5.4 ± 0.2 (1s) (Harbert, 2019) |

**\* Range of single grain AHe dates from this sample**

**[a] The number of grains for which high quality CT data were acquired. Not all grains in this dataset were included in the regressions; see Sect. 4.4.**

### 3.3 Selecting a Representative Crystal Size Distribution

The size distribution of grains analyzed in this study was designed to be representative of the size distribution of apatite grains routinely analyzed for (U-Th)/He dates. We first plotted the maximum width (the larger of the two measured widths; Fig. 1; see also Sect. 4.2) of apatite grains (N = 1061; Fig. 2) analyzed in the CU TraIL over a two-year period. The grains in this dataset were from a variety of sources and were selected and measured by TRaIL staff, TRaIL students, and visitors. Our analysis focused on crystal width because the smaller grain dimension (i.e., the width) is the chief control on alpha ejection due to the long stopping distances of alpha particles. Maximum width was used because for apatite it can be particularly difficult to reliably and accurately measure the minimum width (see Sect. 4.2 and 5.1). These lab analyses were subdivided into small (< 50 μm maximum width), medium (50-100 μm maximum width) and large (>100μm maximum width) size categories (shading in Fig. 2). From the samples described above we then picked suites of apatite crystals for CT with size distributions that were the same as that in the compiled dataset to ensure our analysis covered the full

size range of typically analyzed apatite (Fig. 2). The grains in our initial apatite suite for CT analysis
range in maximum width from 40 to 160 µm.

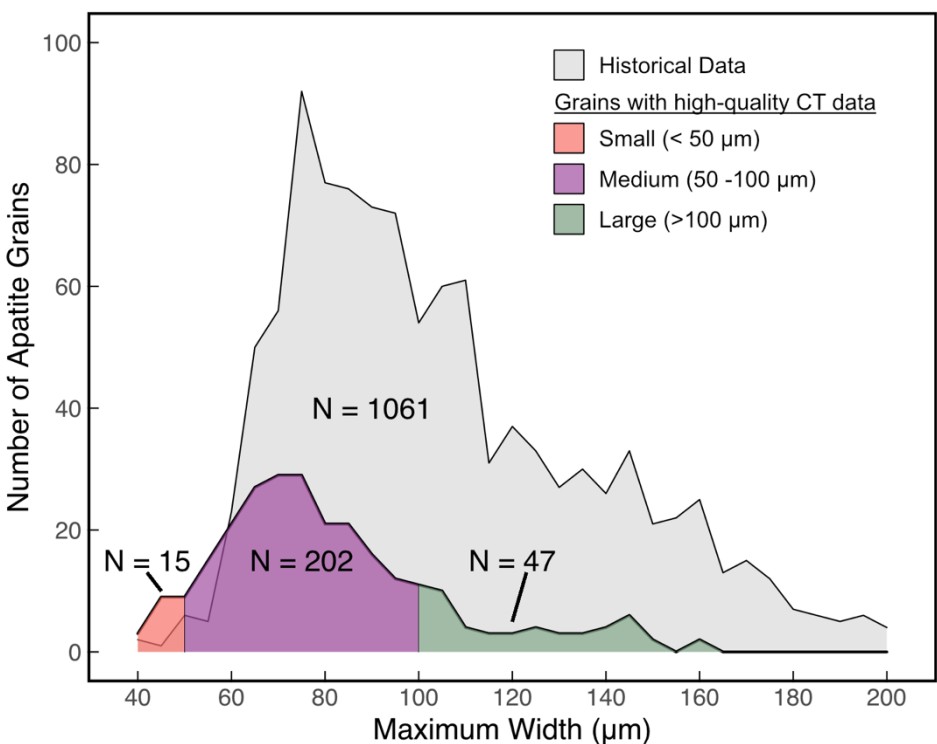

**Figure 2. The distribution of maximum widths of apatite in this study. Light grey depicts 1061 apatite grains dated in the CU TRaIL between 2017-2019. Colored shading illustrates the size distribution of all grains for which we acquired high-quality CT**
**data, with the number of grains in each size category listed. Note that not all grains shown here are included in the final regressions (for example, apatite grains with 3D or 2D $F_T$ values < 0.5 were excluded from the regression analysis).**

### 3.4 Selecting a Morphologically Representative Crystal Suite and Designing the Grain Evaluation Matrix

The morphology of the apatite grains used in this study encompass the spectrum of those regularly
dated by (U-Th)/He. Prior to selecting grains for CT analysis, hundreds of apatite were inspected to gain a sense of the range of grain characteristics. These observations were then used to design a Grain Evaluation Matrix (GEM) (Fig. 3). This was done in part to evaluate whether specific grain qualities are associated with different systematic error or different uncertainty in the geometric parameters. The GEM provides a simple and reproducible method for categorizing the morphologic characteristics of
apatite through which a code (e.g., A1) succinctly describes the morphology of a crystal.

The GEM has two axes (Fig. 3): a "geometric classification" x-axis and a "roughness index" y-axis. Geometry and surface roughness were chosen for the GEM because apatite inspection revealed that

these are the morphological features most likely to contribute to a grain's deviation from the idealized
hexagonal or ellipsoidal geometry used to calculate 2D geometric parameters. In the GEM, geometry is
described as A (hexagonal), B (sub-hexagonal), or C (ellipsoidal), where A and B grains assume a
hexagonal geometry and C grains an ellipsoidal geometry for 2D calculations (Ketcham et al., 2011).
Surface roughness is described as 1 (smooth) or 2 (rough).

Grains with missing terminations are sometimes analyzed by (U-Th)/He, so a subset of grains with one
or two missing terminations was selected for CT analysis. For apatite, grains with missing terminations
are approximately similar in proportion to those in the overall apatite sample suite.

For each apatite GEM category, grains from at least two samples and as many as eight samples were
selected for CT analysis to ensure a range of subtle differences among grain types (Fig. B1). The
number of grains selected for CT analysis in each GEM category was approximately proportional to the
abundance of grains in that category in the entire sample suite. For example, because B1 (sub-
hexagonal, smooth) apatite crystals were more common than C2 (ellipsoidal, rough) crystals in the
apatite suite, more B1 than C2 apatite were analyzed by CT.

Grain roughness (the y-axis of the GEM) was ultimately determined to have no bearing on the
corrections or uncertainties derived in this study. Despite this, the GEM retains this axis because the
GEM is a simple, coherent, and consistent tool for identifying and communicating grain characteristics
that can influence the (U-Th)/He date. Noting the roughness of an apatite grain is useful for evaluating
overall sample quality and can aid in identifying and evaluating dispersion in a (U-Th)/He dataset. The
GEM provides a way to easily report the overall morphologic character and quality of dated apatite
grains. It also is a useful teaching tool to show newcomers to mineral picking the wide variety of
morphologies possible for apatite grains.

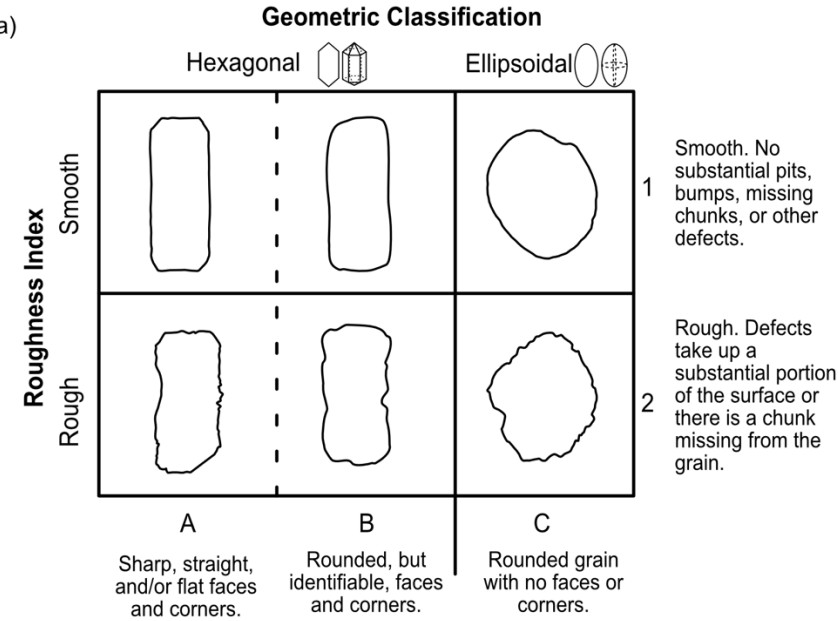

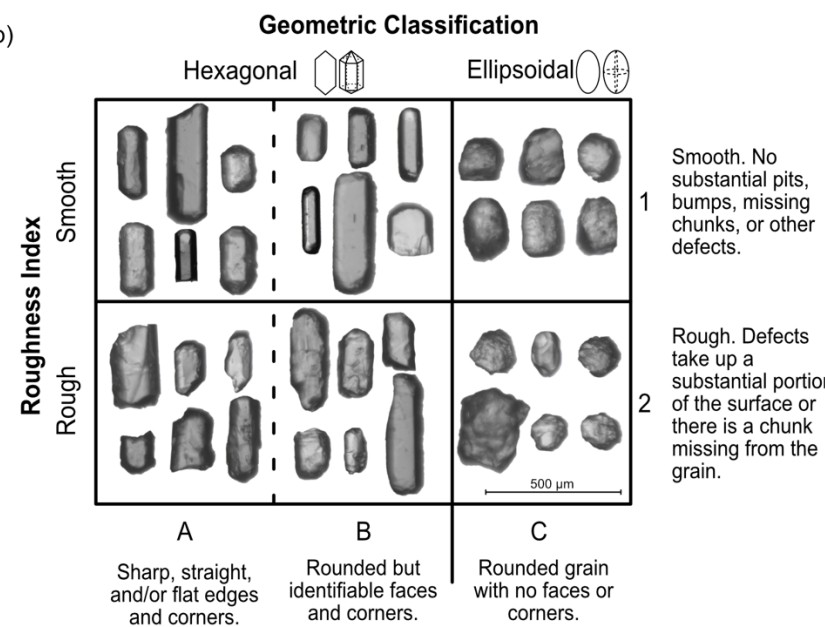

**Figure 3. The apatite Grain Evaluation Matrix (GEM) in (a) schematic form and (b) with images of real grains analyzed in this study. The geometric axis classifies grains as A (hexagonal), B (sub-hexagonal), or C (ellipsoidal). Both A and B apatite grains assume an idealized hexagonal prism geometry while C apatite grains assume an idealized ellipsoidal geometry for 2D calculations (Ketcham et al., 2011). The roughness index classifies grains as 1 (smooth) or 2 (rough). Grains can be described by combining a geometric value and a roughness value (eg. A1, B2).**

## 4 Measurement and Data Reduction Methods

### 4.1 Strategy

The goals of this work are to develop corrections for systematic error and assign appropriate uncertainties to conventional "2D" microscopy estimates of the geometric parameters by comparing 2D values with "3D" values derived from CT data. To do this we first measured our suite of representative apatite crystals using the 2D microscopy approach (Sect. 4.2) and then acquired high-resolution (0.64 μm) CT data for these grains (Sect. 4.3). We then examined the 2D-3D relationships, linearly regressed them to determine corrections depending on grain geometry that make the 2D measurements as close to the 3D values as possible, and calculated uncertainties (Sect. 4.4). This analysis assumes that the 3D values are accurate (Sect 4.3). The final corrections and uncertainties are most appropriate for grains with characteristics like those used in this calibration study, with geometries like those in Figure 3 and microscopy measurements and 2D calculations done as described below. The apatite grains in this work have length/maximum width ratios of 0.8-3.6 and maximum width/minimum width ratios of 1-1.7. $F_T$ uncertainties include only those uncertainties associated with grain geometry and not those due to parent isotope zonation, grain abrasion, or crystal breakage.

### 4.2 Microscopy measurements and 2D calculation methods

Apatite grains were hand-picked under a Leica M165 binocular microscope under 160X magnification. Each grain was photographed on a Leica DMC5400 digital camera, manually measured using either the Leica LAS X or Leica LAS 4.12 software, and assigned a GEM value (Fig. 3). The calibration of the software was checked before, during, and after the measurements using a micrometer. The measurement procedure consisted of first identifying the long axis of the apatite grain parallel to the c-axis, then identifying the apatite's maximum width that is perpendicular to the grain length, acquiring a photograph using the Leica digital camera, and then measuring the length and maximum width using the Leica software (Fig. 4). This was followed by attempting to roll the grain 90°, acquiring another grain photograph, again measuring the long axis using the Leica software to obtain a second length measurement, and estimating and measuring the apatite's "minimum width". Typically, the minimum width that was measured was less than the observed width following grain rotation, because it challenging to efficiently position the grain such that its true minimum width is perfectly visible for measurement in the field of view. This difficulty of rotating and stabilizing the grain for a photograph while the grain is balanced on its minimum width axis makes it difficult to determine and measure the apatite's minimum width accurately. For rounded grains (GEM C, ellipsoidal idealized geometry), the length and widths can be particularly difficult to identify. Our microscopy measurement method is similar to that used in many labs, although a common practice is to acquire only one grain image and thus only a single width measurement (e.g., Cooperdock et al., 2019).

We find a typical 2D measurement uncertainty of 2.8 µm at 1s standard deviation. This was determined based on repeat measurements by 3 individuals of 258 apatite grains using the same images and software for each grain. Each individual measured both lengths and the maximum width of each grain, for a total of 774 measurements per person. The 1s sample standard deviation for each grain dimension was calculated, with an average standard deviation of 2.8µm.


a) Hexagonal

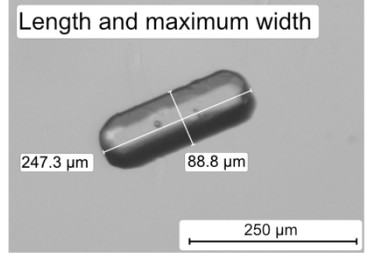

b) Ellipsoidal

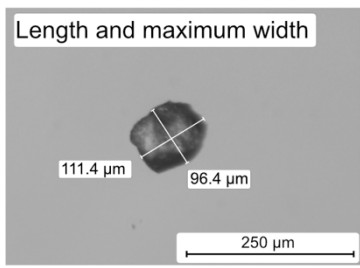

**Figure 4. Photomicrographs of a) hexagonal and b) ellipsoidal apatite grains showing how each grain's length and maximum width were measured for the 2D microscopy measurements. After these two measurements were complete, the grain was then**
**rolled 90° onto its side, another photomicrograph of the grain was acquired, and a second length and the minimum width measurements were acquired, with the latter aimed at closely approximating the $W_{min}$ axis shown in Figure 1.**

The 2D V values and the isotope-specific $F_T$ values were calculated assuming the idealized geometries and equations in Ketcham et al. (2011). $R_{FT}$ values were calculated using the equations in Cooperdock et al. (2019). We used the mean stopping distances for [238]U, [235]U, [232]Th, and [147]Sm from Ketcham et al.
(2011). The $F_T$ calculations of Ketcham et al. (2011) assume that every surface is an ejection surface. All equations are listed in Appendix A. A hexagonal geometry was used for all A and B (hexagonal and sub-hexagonal) grains, while an ellipsoidal geometry was used for all C grains. For each apatite, we calculated the $R_{FT}$ value by assuming an apatite Th/U ratio of 1.94 and no contribution from Sm, where the Th/U ratio is the average of the TRaIL apatite sample historical data (N = 1061 grains) shown in
Fig. 2. We made this assumption because the $R_{FT}$ depends on the proportion of each parent isotope contributing to [4]He production, and we do not have parent isotope values for the grains analyzed by CT in this study.

The 2D V, $F_T$, and $R_{FT}$ values were computed using two different combinations of measurements: 1) using each apatite grain's length, maximum width, and minimum width measurements, and 2) using each apatite grain's length and only the maximum width value by assuming that the minimum width is equal to the maximum width (Fig. 1; Appendix A). Our favored calibration ultimately uses the second approach owing to the difficulty of measuring the minimum width accurately, as discussed further below (Sect. 5.1, 6.1).

### 4.3 Nano-computed tomography and 3D calculation methods

After 2D measurements, apatite grains were mounted for CT. Crystals were mounted in an ~600 x 600 µm area on a thin, 2000 µm wide plastic disc that was hole-punched from a plastic sheet protector and then covered with double sided tape (Fig. 5). Each plastic disc was constructed with a 0.025 mm diameter wire running down the center to act as a point of orientation to aid in the identification of grains post-scan. It was later discovered that the high-density wire created challenges for data reduction, so this approach is not recommended for future studies. Each plastic disc held 4-10 grains and 5-6 discs were stacked vertically to create a mount (Fig. 5). Mounts were secured by a thin layer of parafilm, attached to a 1-2 mm thick rubber cylinder for stabilization, and then glued to the top of a flat-head pin (Fig. 5).

Each mount was scanned on a Zeiss Xradia 520 Versa X-ray Microscope in the University of Colorado Boulder Materials Instrumentation and Multimodal Imaging Core (MIMIC) Facility. Scanning parameters were optimized to reduce noise and scanning artifacts during test scans of the first mount. Scanning parameters were kept constant for subsequent mounts. All mounts were scanned with the 20X objective at relatively low power and voltages with small distances between the mount, source, and detector, which allowed for high resolution (0.64 µm). Table B1 reports the scan parameters.

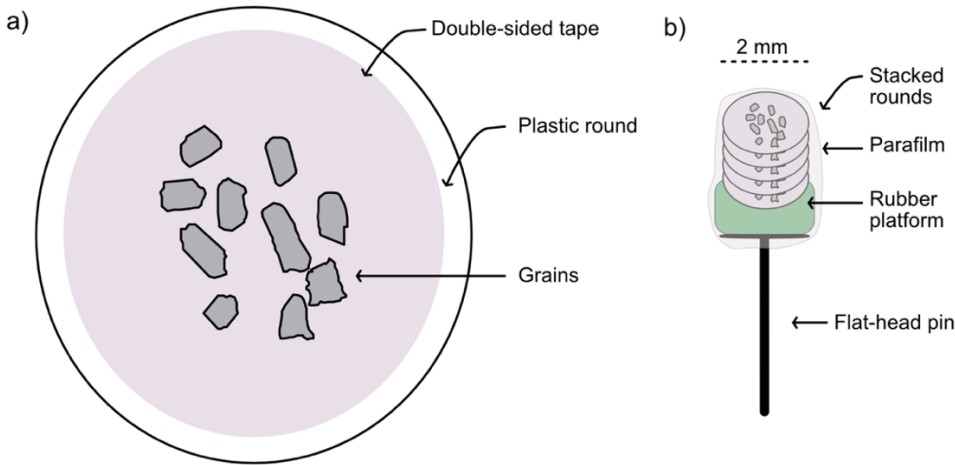


**Figure 5. Schematic showing (a) an individual plastic round and (b) a final grain mount for CT analysis. The wire is not shown because it should not be included in future studies. Grains are placed onto a ~2 mm wide sturdy plastic disc (hole punched from a plastic sheet protector) covered with double-sided tape. Each plastic round can hold between 4-10 grains. Rounds are stacked on top of each other and placed on a rubber platform cut from old test tube stoppers, which is glued to a flathead pin and covered**

**with Parafilm.**

Raw CT data were imported into Blob3D (Ketcham, 2005; freely distributed software) to calculate the dimensions, V, surface area, and isotope-specific $F_T$ values for each grain. First, the grains were segmented, or separated, from the matrix, noise, and other grains, such that each grain was a separate

'blob'. Segmentation was done with Dragonfly software, Version 2020.2 for Windows (Object Research Systems, v.2020.2) due to the complex nature of the artifacts arising from the use of the wire. After segmentation, the 3D V, surface area, and $F_T$ values were calculated by Blob3D. Blob3D calculates grain dimensions by first identifying the shortest caliper dimensions (Box C), then measuring the shortest dimension orthogonal to it (Box B), and finally measuring the dimension orthogonal to both

(Box A) (Ketcham, 2005; Cooperdock et al., 2019). These dimensions generally correspond to the minimum width (Box C), the maximum width (Box B) and the length (Box A) of a regularly shaped apatite (Cooperdock et al., 2019). Blob3D calculates V by counting the number of voxels (3D-pixels) in the segmented object and multiplying that number by the volume of each voxel. Blob3D calculates surface area by summing the faces of the isosurface surrounding the grain voxels and then smoothing it

to reduce the effects of pixelation caused by the cubic voxels (Ketcham, 2005; Cooperdock et al., 2019). Blob3D calculates the $^{238}U$, $^{235}U$, $^{232}Th$, and $^{147}Sm$ $F_T$ corrections using a Monte Carlo approach that randomizes the starting location of an alpha particle within the selected volume of an object. The direction of ejection of the alpha particle is calculated via uniform sampling (Ketcham and Ryan, 2004). Blob3D uses stopping distances as reported in Ketcham et al. (2011) and assumes that ejection occurred

across all surfaces. As for 2D $R_{FT}$ values, we calculated 3D $R_{FT}$ values using the equations of Cooperdock et al. (2019) and assuming a Th/U ratio of 1.94 based on TRaIL apatite sample historical data.

In order to confirm our assumption that the CT measurements are representative of the "real" grain
measurements, we assumed a ± 1% uncertainty in our CT measurements, based on preventative
maintenance measurements performed by the MIMIC lab and technicians, and performed simulations in
Blob3D by varying the voxel size similar to those done in Cooperdock et al. (2019). Like Cooperdock et
al. (2019), we find that uncertainties in the CT data translate to negligible differences in the relevant
values output by Blob3D and are vanishingly small compared to the uncertainties in the 2D
measurements.

Some apatite grains were removed from the final dataset owing to issues during CT scanning or
subsequent data processing. Due to the use of the 20X objective for high resolution, many of the
original 400 grains were lost because the edges of grains were 'cut off' during scanning. Additionally,
the high-density wire in the apatite mounts introduced challenges for data reduction, such as 3D models
that had large holes or complex surface artifacts. The final dataset after removal of the grains with
problematic analytical results consists of 264 apatite grains with high-quality CT data.

### 4.4 Statistical comparison of 2D and 3D values

The first step in our 2D-microscopy vs. 3D-CT data comparison was to generate scatter plots of 3D vs.
2D data for all 264 apatite grains in our dataset. Figure 6 shows these plots for length, maximum width,
and minimum width. Figure 7 includes these plots for V, isotope-specific $F_T$, and $R_{FT}$ values. In Figure
7, we show only the isotope-specific $^{238}U$ $F_T$ value for illustrative purposes because $^{238}U$ dominates the
$^4He$ production budget. However, we plotted and regressed the data for the $^{235}U$, $^{232}Th$, and $^{147}Sm$
isotope-specific $F_T$ values in the same manner as for the $^{238}U$ $F_T$ value and include those plots in Figure
C1. We did not examine surface area separately because although it is used together with volume to
determine the $F_T$ value, it is not alone used to calculate any other geometric parameter (unlike volume,
which is used to calculate concentrations).

Our next step was to carry out regressions of the 3D vs. 2D data for V, isotope-specific $F_T$, and $R_{FT}$
values. On the 3D versus 2D plots, if the data fall on the 1:1 line (bold black line), then no correction
for systematic error is needed for the 2D data because the 2D data are in agreement with the 3D data. If
the data fall off the 1:1 line, then the correction desired for the 2D data can be viewed as the offset of
the data and its linear regression line from the 1:1 line. To determine corrections for systematic error,
ordinary least squares linear regression with the intercept fixed at the origin was used. We explored
several regression approaches, but ultimately chose an unweighted approach because the scatter of the
2D data that we wish to characterize includes both the uncertainty on the grain length and width
measurements and other factors such as surface roughness and deviation from the assumed idealized
grain geometry. We also explored fixing versus not fixing the y-intercept at (0,0). Here we present only
the results of regressions with the y-intercept fixed at 0, because the unconstrained regressions generally
yield intercepts within uncertainty of 0 and we would expect that if 2D measurement of any parameter
was 0, then the 3D value would also be 0.

We ultimately excluded from the regressions the apatite (N = 27) with 3D $F_T$ values <0.5, which are
grains smaller than those typically analyzed by (U-Th)/He. This was done to avoid biasing the corrections and uncertainties with data for grains that are not representative of regularly dated apatite. This exclusion resulted in the elimination of all "small"-sized grains with <50 μm maximum width from the regressions. These small grains are characterized by greater differences between 2D and 3D values and higher scatter than the medium- and large- sized grains in our dataset, as shown by the grey points
in Figure 7. The final regressed dataset has 237 apatite grains.

To evaluate if different groups of grains have statistically different slopes (and thus should have different corrections applied to them) we used Tukey's test (Table C1). Separate linear regressions were done for grains that use different geometric assumptions, so hexagonal apatite (A and B grains in Figure
3) were regressed separately from ellipsoidal apatite (C grains in Fig. 3). The slopes for the linear regressions of these two groups are statistically distinguishable, justifying their separation by geometry. Linear regressions were also done by grouping by surface roughness (1 vs. 2 on the GEM, Fig. 3) and size (medium, large). The linear regression slopes for these different categories are each statistically indistinguishable, indicating it is reasonable to only group the data by geometry for all parameters
(Table C1).

The uncertainty for each 2D geometric parameter is the scatter of the points about the regression line. To determine the uncertainty of each 2D parameter, we calculated the 1s standard deviation of the residual values of all points from the regression line. This is shown on Figure 7 as plots of residual
percent difference versus maximum width for each parameter. To assess if physical parameters (e.g., roughness, size) are associated with patterns in these residuals, we compared the standard deviations for different groups of physical variables (Table C2).

The correlation of isotope-specific $F_T$ uncertainties was also evaluated because we expect them to be
highly correlated (Martin et al., 2023). The correlation coefficient between each isotope-specific $F_T$ was calculated using Pearson's r.

## 5 Results: Corrections and Uncertainties

### 5.1 Comparison of grain dimensions from 2D microscopy and 3D CT data

For apatite dimension data, the 3D versus 2D scatter plots illustrate that the 2D values accurately
measure the length (Box A) and the maximum width (Box B), with average 3D/2D values of 1.0 and 0.99 and average differences of 5%, respectively (Fig 6a-b). Outliers are due to oddly shaped or fragmented grains, which can be measured inaccurately by the procedure used by Blob3D (Cooperdock et al., 2019). However, we find that the third dimension, the minimum width, is more difficult to measure accurately via microscopy (Fig. 6c). Our 2D minimum width measurements consistently
underestimate the 3D Box C measurements with a large amount of scatter; the average 3D/2D value is 1.09 with an average absolute difference of 13%. This inaccuracy and high uncertainty are attributable

to the practical challenges associated with photographing an apatite crystal in the proper orientation for minimum width measurement (Sect. 4.2). The systematic 2D underestimation of the minimum width is because the analyst was aware that the observed width of the grain after attempting to roll it 90° from the maximum width position (Sect. 4.2) was larger than the apatite's actual minimum width, and then overcompensated for this fact by acquiring a measurement that was not only smaller than the observed width but also inadvertently smaller than the true minimum width.

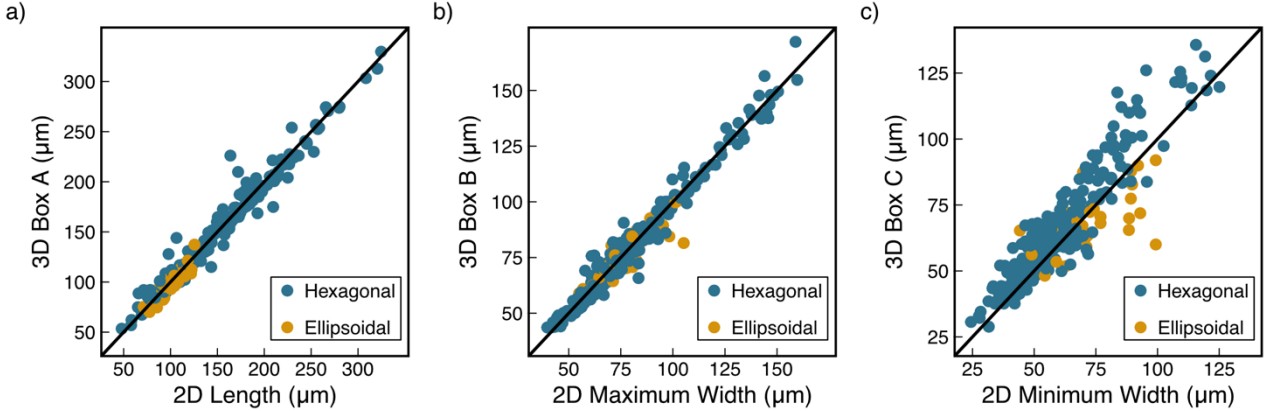

**Figure 6. Scatter plots of 3D vs. 2D data (N = 264) for grain dimension measurements. a) 3D Box A vs. 2D length measurement, b) 3D Box B vs. 2D maximum width measurement, and c) 3D Box C vs. 2D minimum width measurement. The bold black line is the 1:1 line. Note that the 2D minimum width data have greater scatter and systematically underestimate the corresponding 3D Box C measurement.**

## 5.2 Corrections for systematic error

The 3D versus 2D scatter plots for V, $F_T$, and $R_{FT}$ (Figure 7a-c) using the maximum width for both width values for 2D calculations all show data that systematically plot below the 1:1 line (bold black line), indicating that for all parameters the 2D values overestimate the "true" 3D values. The 2D data can be corrected for their systematic overestimation of the 3D data by multiplying the 2D data by the slope of the 3D vs. 2D data, so that the 2D data are centered around the 1:1 line, thereby "correcting" them. As noted in Sect. 4.4, regressions of the 3D vs. 2D data are separated by geometry because the regressions of hexagonal and ellipsoidal grains yield statistically distinguishable slopes.

The corrections for systematic error for apatite V, $F_T$, and $R_{FT}$ are summarized in Table 2. For all parameters, the magnitude of the correction is smaller for hexagonal grains than for ellipsoidal grains. For example, for V, the slope of the regression line is 0.83 for hexagonal grains and 0.74 for ellipsoidal grains. This means that the volumes estimated by microscopy measurements typically overestimate the true grain volume by 17% for hexagonal grains, and by 26% for ellipsoidal grains. For [238]$F_T$, the corrections are substantially smaller, with values of 0.97 and 0.92 for hexagonal and ellipsoidal grains. For $R_{FT}$, the corrections are 0.93 and 0.85 for hexagonal and ellipsoidal grains.

Figure D1a-c includes 3D versus 2D scatter plots for V, $F_T$, and $R_{FT}$ using both the maximum and minimum width values for 2D calculations, with the associated corrections for systematic error summarized in Table D1. In this case, for hexagonal grains, all data systematically plot above the 1:1 line (bold black line), indicating that the 2D values consistently underestimate the "true" 3D values (Figure D1a-c). The corrections for systematic error are systematically larger for all parameters than the corrections using only the maximum width (Table 2). For example, for hexagonal grains, V, $F_T$, and $R_{FT}$ are underestimated by 27%, 8%, and 15%, respectively when using both widths, compared with an overestimation of 17%, 3%, and 7% when using only the maximum width. The underestimation of 2D values when using both widths is due to microscopy measurements that systematically underestimate minimum width values (see Sect. 5.1). For ellipsoidal grains, using both widths causes 2D values to overestimate the 3D values (the 2D data plot below the 1:1 line in Figure D1a-c), however the magnitude of these corrections are the same or slightly smaller than when using only the maximum width for 2D calculations (Table 2, D1).

## 5.3 Uncertainties

The uncertainties for V, $F_T$, and $R_{FT}$ are derived from the scatterplots of the percent difference in the residuals versus maximum width where the bold black line represents no difference between the 2D and 3D data (Figure 7d-f, for the analysis using the maximum width only for 2D calculations). The uncertainties are grouped by geometry for all parameters, because the residuals are derived from the regression lines, which group data in this way. The standard deviation of the percent difference in the residuals of each group is the uncertainty on each parameter, reported in Table 2 at 1s. A single uncertainty is reported for ellipsoidal apatite grains for all parameters due to the relatively small number of ellipsoidal grains in the dataset (N = 36). However, for hexagonal grains, the data population (N= 201) is large enough that we explored surface roughness and grain size as potential grouping variables. We did not find a consistent, substantial relationship between surface roughness and uncertainty in the data (Table C2). However, for grain size, the $^{238}F_T$ uncertainty for medium-sized (maximum width 50-100 μm) hexagonal apatite is greater than for large-sized (maximum width > 100 μm) hexagonal apatite. As described below, this pattern is sensible, so we report two uncertainties for the isotope-specific $F_T$ values of hexagonal grains based on size.

For all parameters, the uncertainty for hexagonal grains is smaller than the uncertainty for ellipsoidal grains (Table 2). For V, the uncertainty is 20% for hexagonal grains and 23% for ellipsoidal grains of all sizes. For $^{238}F_T$, the uncertainties are 3% and 2% for medium and large hexagonal grains, respectively, and 5% for all ellipsoidal crystals. For $R_{FT}$, the uncertainty is 6% for hexagonal grains and 10% for all ellipsoidal grains of all sizes.

As anticipated, the isotope-specific $F_T$ uncertainties are highly correlated, yielding correlation coefficients of 0.972-0.999. For this reason, below we assume fully correlated uncertainties of 1 for $F_T$ uncertainty propagation into the corrected date.

Figure D1d-f includes the relevant scatterplots for determining uncertainties on V, $F_T$, and $R_{FT}$ using both the maximum and minimum width values for the 2D calculations, with the derived uncertainty values listed in Table D1. For hexagonal grains, the uncertainties are consistently larger for all parameters when using both widths rather than only the maximum width for 2D calculations. For ellipsoidal grains, the uncertainties are the same or larger when using both widths instead of only the maximum width (Table 2, D1).

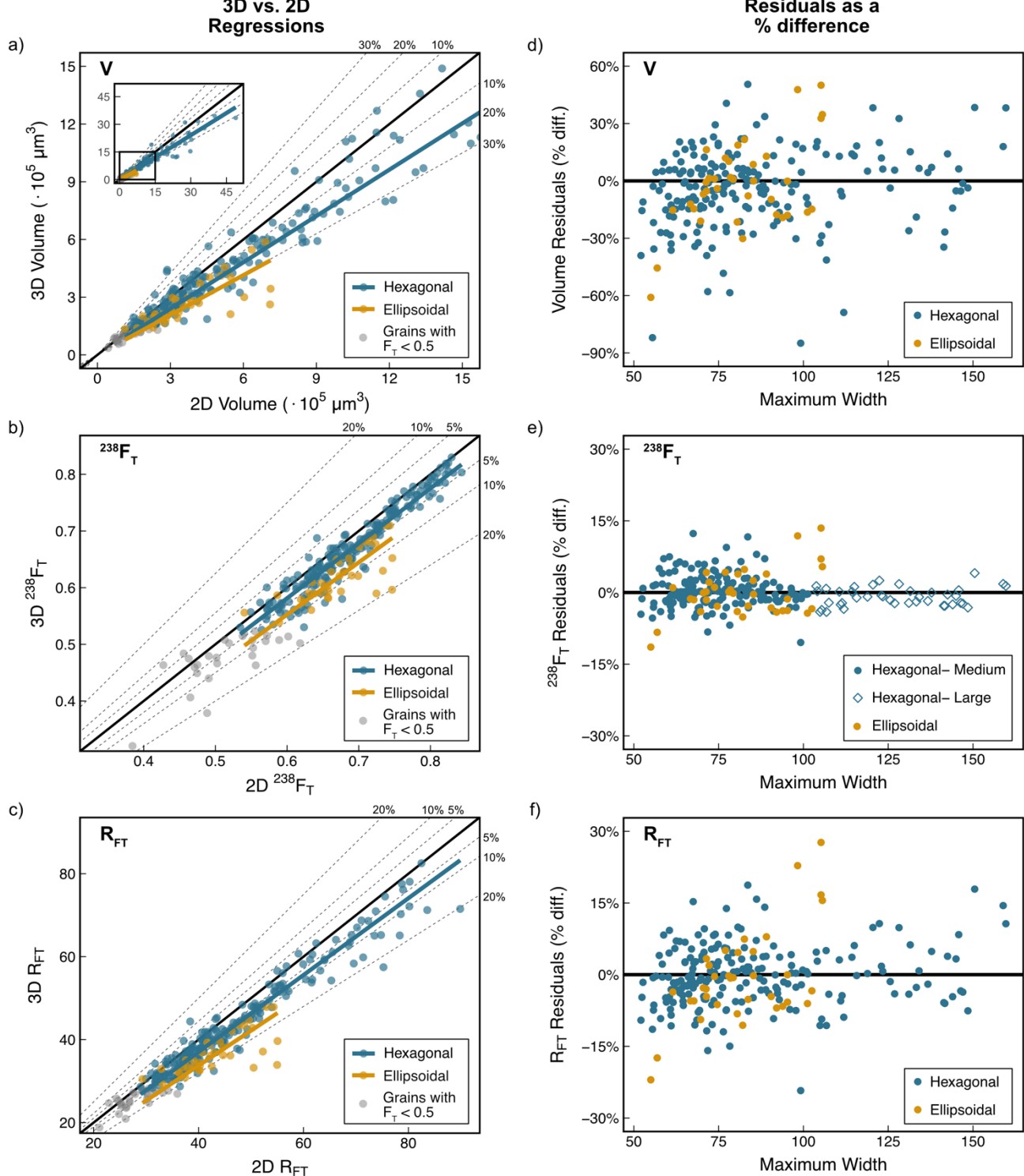

**Figure 7. Plots illustrating how the corrections for systematic error and how uncertainties were determined for V, $F_T$, and $R_{FT}$. Scatter plots of 3D vs. 2D data (N = 264) with regression lines and data distinguished by geometry for (a) V, (b) $^{238}F_T$, and (c) $R_{FT}$. 2D data in these plots were calculated using the maximum width for both width values (see Appendix A). Grains with $F_T < 0.5$ were excluded from the regressions but are included on the plots in light grey. A total of 237 apatite grains are in the regressed dataset. The bold black line is the 1:1 line and the dashed lines mark the percent difference from the 1:1 line. Note that for all**
**regressions, the regression line falls below the 1:1 line, indicating that the 2D-microscopy data overestimate the 3D-CT data. The 2D data can be corrected for systematic error by multiplying the 2D data by the slope of the regression. Plots of the difference of each 2D value from the regression line (i.e., the residual) as a percent difference vs. maximum width with data distinguished by geometry for (d) V, (e) $^{238}F_T$, and (f) $R_{FT}$. For $^{238}F_T$ the hexagonal grains are additionally split by medium (50-100 μm maximum width) vs. large (> 100 μm maximum width) size. The bold black line is 0% difference. Note the larger y-axis scale for V as**
**compared with $^{238}F_T$ and $R_{FT}$, reflecting the greater uncertainty of V. The standard deviation of the % difference in the residuals of each group is the uncertainty on the parameter.**

**Table 2. Corrections and uncertainties (1s) for all geometric parameters.**

### Volume

| Geometry | Correction[a] | % Uncert.[b] (1s) for apatite grains of all sizes |
|---|---|---|
| Volume | | |
| Hex. | 0.83 | 20% |
| Ellip. | 0.74 | 23% |

### Isotope-specific $F_T$ values

| Geometry | Correction | % Uncert. (1s) for medium-sized[c] apatite grains | % Uncert. (1s) for large-sized[d] apatite grains |
|---|---|---|---|
| $^{238}F_T$ | | | |
| Hex. | 0.97 | 3% | 2% |
| Ellip. | 0.92 | 5% | 5% |
| $^{235}F_T$ | | | |
| Hex. | 0.96 | 4% | 2% |
| Ellip. | 0.91 | 6% | 6% |
| $^{232}F_T$ | | | |
| Hex. | 0.96 | 4% | 2% |
| Ellip. | 0.91 | 6% | 6% |
| $^{147}F_T$ | | | |
| Hex. | 0.99 | 1% | 1% |
| Ellip. | 0.97 | 1% | 1% |

### $R_{FT}$

| Geometry | Correction | % Uncert. (1s) for apatite grains of all sizes |
|---|---|---|
| $R_{FT}$ | | |
| Hex. | 0.93 | 6% |
| Ellip. | 0.85 | 10% |

 [a] The correction value is the slope of the 3D vs. 2D regression line for each parameter in Figures 7a-7c.

[b] The uncertainty is the scatter of the 2D data about each regression line in Fig. 7a-c, calculated as the 1s standard deviation of the % difference of each 2D value from the regression line (Fig. 7d-f).

[c] "Medium-sized" apatite have maximum widths of 50-100 μm.

[d] "Large-sized" apatite have maximum widths of >100 μm.

## 6 Discussion

### 6.1. Measurements and grain characteristics that influence the accuracy and precision of 2D geometric data

The goal of this study was to develop a simple method for correcting for systematic error and for assigning uncertainties to geometric parameters estimated from microscopy measurements for the full spectrum of apatite grains that are regularly analyzed by (U-Th)/He. Thus, the corrections for systematic error are intended to improve the accuracy of the V, $F_T$, and $R_{FT}$ values derived from 2D data. The uncertainties are aimed at appropriately representing the reproducibility or precision of these geometric parameters. Accomplishing this required determining the measurements and grain characteristics that most affect the accuracy and precision of the 2D data.

Whether only the length and maximum width, or the length, maximum width, and minimum width are used for calculating the 2D geometric parameters influences both the magnitude of the correction for systematic error and the uncertainties (Sect. 5). We recommend using the maximum width only rather than both the maximum and minimum widths for 2D geometric parameter calculations for several reasons. First, the length and maximum width are the two most accurately and reproducibly measured dimensions; it is difficult to efficiently and reliably measure the apatite's minimum width (Fig. 6c; Sect. 5.1). Second, no excess correction or uncertainty is introduced by measuring and using only the maximum width rather than both widths. In fact, for most apatite grains, the corrections and uncertainties are higher when using both width measurements (see Sect. 5), due to the inaccuracy and scatter of the 2D minimum width data. Third, it is common practice in many labs to acquire only one grain image and measure only an apatite's maximum width (Cooperdock et al., 2019), such that this set of corrections and uncertainties may be more widely useful. This may be especially true for retroactive application to published data. Finally, time is saved by not acquiring a second set of measurements at no detriment to the data quality. The rest of our discussion below is focused entirely on these outcomes that use only the length and maximum width in the 2D calculations.

We find that the first-order grain morphology is the grain characteristic that most strongly influences the magnitude of the systematic error on 2D geometric data. For example, whether apatite grains are hexagonal or sub-hexagonal (A or B on the GEM) versus ellipsoidal (C on the GEM), dictates the choice of a hexagonal or ellipsoidal idealized geometry. This in turn determines the magnitude of the

correction required to make the geometric parameters calculated from the microscopy data accurate (e.g., for $^{238}F_T$ a 0.97 correction for hexagonal grains vs. a 0.92 correction for ellipsoidal grains).

Our results show that the uncertainty in the 2D geometric parameters is controlled primarily by the grain geometry, and for $F_T$, secondarily by the grain size. Uncertainties on hexagonal grains are consistently smaller than those for ellipsoidal grains (Table 2). For example, for V, uncertainties are 20% and 23% for hexagonal and ellipsoidal grains, respectively. For $R_{FT}$, the uncertainties on hexagonal grains (6%) are again smaller than for ellipsoidal grains (10%). For $^{238}F_T$, grain size exerts additional influence on the uncertainty of hexagonal grains, with uncertainties of 3% and 2% for grains with maximum widths of 50-100 µm and > 100 µm, respectively, compared with an uncertainty of 5% for ellipsoidal grains of all sizes. The influence of size on the $F_T$ uncertainty is not surprising given that the effect of the uncertainty in grain measurements is proportionately larger for smaller grains. This pattern is consistent with early work that estimated $F_T$ uncertainty increased with decreasing grain size (Ehlers and Farley, 2003).

## 6.2 Overestimation of the 3D geometric parameter values by the 2D microscopy method

### 6.2.1. Overview

In this study, all values calculated from the 2D microscopy measurements overestimate the real 3D values (when using length and maximum width for 2D geometric parameter calculations, as discussed in the previous section). This overestimation is true regardless of grain size, morphology, and other grain characteristics. Compared with past work (Herman et al., 2007; Evans et al., 2008; Glotzbach et al., 2019; Cooperdock et al., 2019), in this study we analyzed more apatite (237 compared with 4-109) and at a higher CT resolution (0.64 µm compared with 1.2 -6.3 µm). We also deliberately included the full variety of grain morphologies across a range of grain sizes from samples of variable age and lithology, so we have confidence that the results are applicable to the spectrum of routinely analyzed apatite.

As explained in Sect. 4.4, the corrections and uncertainties discussed above and reported in Table 2 are calculated from the regressions and are computed in this way because the objective of our work is to systematically correct real 2D data and routinely apply the associated uncertainty to them. However, previous studies, which did not have these same goals in mind, reported the average 3D/2D value and its 1s uncertainty as a measure of systematic error, and reported the average absolute percent difference between the 2D and 3D data and its 1s uncertainty as a measure of the uncertainty of each parameter. To directly compare our results to this past work, in Table 3 we also report our results in this way. This table directly follows the structure of Table 3 in Cooperdock et al. (2019). In our Table 3, we report values for our entire dataset, as well as subdivided by hexagonal and ellipsoidal geometry. However, for simplicity, we use only the average values for our whole dataset in the discussion below.

We place our results in the context of those of Cooperdock et al. (2019) and Glotzbach et al. (2019) because these two studies directly compared 2D microscopy with 3D CT values for a moderate to large

suite of apatite crystals. Cooperdock et al. (2019) characterized 109 hexagonal to sub-hexagonal apatite grains (A1 and B1 in our GEM) by CT (5 µm resolution) and calculated 2D parameters using the length and maximum width only. Glotzbach et al. (2019) analyzed 24 apatite crystals (1.2 µm CT resolution) with a wider range of characteristics (rounded through euhedral morphologies) and calculated 2D parameters using measurements of the length, maximum width, and minimum width. Although Evans et

al. (2008) also carried out a study of this kind and was the first to do this type of comparison, that work included only four apatite crystals (3.8 µm CT resolution). Herman et al. (2007) used CT to derive geometric parameter data for 11 detrital apatite grains (6.3 µm CT resolution) but did not compare the results with 2D microcopy estimates for the same grains.

**Table 3. 2D microscopy and 3D CT data comparison for this and previous studies[a]**

| This Study: 237 apatite grains; CT resolution: 0.64 μm | | | | |
|---|---|---|---|---|
| | avg. 3D/2D[b] | 1s | abs. avg. % diff.[c] | 1s |
| **All data: 237 grains** | | | | |
| Volume | 0.85 | 0.17 | 19 | 13 |
| $^{238}F_T$ | 0.96 | 0.04 | 4 | 4 |
| $R_{FT}$ | 0.92 | 0.07 | 8 | 6 |
| Length/Box A | 1 | 0.07 | 5 | 6 |
| $W_{max}$/Box B | 0.99 | 0.06 | 5 | 4 |
| $W_{min}$/Box C | 1.09 | 0.14 | 13 | 10 |
| **Hexagonal apatite: 201 grains** | | | | |
| Volume | 0.87 | 0.17 | 18 | 12 |
| $^{238}F_T$ | 0.97 | 0.03 | 4 | 3 |
| $R_{FT}$ | 0.93 | 0.06 | 7 | 5 |
| Length/Box A | 1.01 | 0.07 | 5 | 6 |
| $W_{max}$/Box B | 1 | 0.06 | 4 | 4 |
| $W_{min}$/Box C | 1.11 | 0.12 | 13 | 10 |
| **Ellipsoidal apatite: 36 grains** | | | | |
| Volume | 0.75 | 0.17 | 26 | 15 |
| $^{238}F_T$ | 0.92 | 0.05 | 8 | 4 |
| $R_{FT}$ | 0.86 | 0.08 | 15 | 8 |
| Length/Box A | 0.98 | 0.06 | 6 | 3 |
| $W_{max}$/Box B | 0.97 | 0.07 | 6 | 5 |
| $W_{min}$/Box C | 0.97 | 0.16 | 12 | 11 |

| Previous Studies | | | | |
|---|---|---|---|---|
| | avg. 3D/2D | 1s | abs. avg. % diff. | 1s |
| **Cooperdock et al. (2019): 108 apatite grains; CT resolution: 4-5 μm** | | | | |
| Volume | 0.82 | 0.22 | 23 | 16 |
| $^{238}F_T$ | 1.01 | 0.02 | 2 | 2 |
| $R_{FT}$ | 1.02 | 0.07 | 5 | 5 |
| Length/Box A | 0.98 | 0.1 | 4 | 6 |
| $W_{max}$/Box B | 1.03 | 0.07 | 16 | 8 |
| $W_{min}$/Box C | N/A[d] | N/A | N/A | N/A |
| **Glotzbach et al. (2019): 24 apatite grains; CT resolution: 1.2 μm** | | | | |
| Volume | 1.04 | 0.2 | 15 | 13 |
| $^{238}F_T$ | 0.99 | 0.02 | 2 | 2 |
| $R_{SV}$[e] | 0.93 | 0.06 | 8 | 5 |

[a] Directly follows the structure of Table 3 reported in Cooperdock et al. (2019) to facilitate comparison with previous studies.

[b] avg. 3D/2D is the average of all 3D/2D values in each study

[c] abs. avg. % diff. Is the average absolute percent difference between the 2D and 3D data. We used the formula $\left(\frac{|2D-3D|}{2D}\right) \times 100$ to calculate the percent difference for consistency with Cooperdock et al. (2019).

[d] N/A is not available

[e] Glotzbach et al. (2019) reports $R_{SV}$ rather than $R_{FT}$.

### 6.2.2. Volume

Of the geometric parameters evaluated in this study, V shows the greatest overestimate of 2D relative to 3D values (2D value corrections of 0.83 and 0.74 depending on geometry) and the greatest data scatter (20% and 23%) (based on the data regressions, Table 2). If we instead report our outcomes as the average 3D/2D value and the average absolute % difference, we obtain values of 0.85 and 19% for all grains (Table 3). This result is generally consistent with those of previous work. Cooperdock et al., (2019) also found a V overestimate with an average 3D/2D value of 0.82 and an average difference of 23%. Glotzbach et al. (2019) found no systematic over- or underestimate in volume (avg 3D/2D = 1.04), partly attributable to their use of all three dimensions (3D-He method) in their 2D parameter calculations, and they report a similar magnitude of variation (15%).

### 6.2.3. $F_T$

For $F_T$, our 2D values overestimate the 3D values. The isotope-specific $^{238}F_T$ has a 2D correction value of 0.97 for hexagonal grains and 0.92 for ellipsoidal grains, with uncertainties of 2-5% depending on geometry and size (based on the regressions, Table 2). The corrections and uncertainties for the other isotope-specific $F_T$ values vary from 0.99 to 0.91 and 1-6%, respectively, (again depending on grain geometry and size, Table 2), but we focus on the $^{238}F_T$ value here because it dominates the $^4$He production. Our average 3D/2D value for $^{238}F_T$ is 0.96, with an average difference of 4% (Table 3). This outcome is similar to that of Glotzbach et al. (2019) (avg. 3D/2D = 0.99; avg. abs. diff. = 2%). In contrast, Cooperdock et al. (2019), report 2D values that slightly underestimate the 3D $F_T$ values (average 3D/2D = 1.01), but with a comparable magnitude of scatter (2%). This may be due, in part, to their grain selection, which focused mainly on high quality, hexagonal apatite grains.

### 6.2.4. $R_{FT}$

For $R_{FT}$, we found that 2D measurements were systematically larger than 3D measurements (2D correction values of 0.93 and 0.85), with uncertainties of 6-10% depending on geometry (based on the regressions, Table 2). Our average 3D/2D value for $R_{FT}$ is 0.92, with an average difference of 8% (Table 3). Glotzbach et al. (2019) reports $R_{SV}$ (the equivalent sphere with the same surface area to volume ratio as the grain) rather than $R_{FT}$, but these values typically have negligible difference. Their dataset yields $R_{SV}$ outcomes nearly identical to our $R_{FT}$ results (avg. 3D/2D = 0.93; avg. abs. diff. = 8%). In contrast, Cooperdock et al. (2019) found an average 3D/2D value of 1.02 and an average

difference of 5% (Table 3). Their underestimation of $R_{FT}$ by 2D measurements is expected given the systematic underestimation they report for $F_T$.

### 6.3 Implications: How much do the corrections and geometric uncertainties matter?

**6.3.1 Overview**

To determine how much the corrections and geometric uncertainties (Table 2) affect the values and uncertainties on real (U-Th)/He dates and other key parameters, we apply our corrections and uncertainties to the V, $F_T$, and $R_{FT}$ values of a subset of representative apatite grains from three samples (N = 24) that were used in this study and that were previously dated in the CU TRaIL (Table E1). This
apatite suite includes both hexagonal and ellipsoidal grains with a range of sizes. We then use the corrected V and isotope-specific $F_T$ values to calculate the parameters derived from them—mass, eU, and the corrected (U-Th)/He date—and propagate the geometric uncertainties on V and $F_T$ into the uncertainties of the derived values. Below, we then compare the "Geometric Correction Method" (GCM) values and uncertainties on all parameters with their "2D" uncorrected counterparts (Sect. 6.3.2-
6.3.5), generate corrected apatite (U-Th)/He (AHe) date vs. eU plots using both the GCM and 2D values (Fig. 8), and consider the broader implications of these outcomes for interpretation of AHe data (Sect. 6.3.6).

Table 4 summarizes the average GCM/2D values for this example dataset, as well as how much the
700 uncertainty on each parameter increases owing to the inclusion of geometric uncertainties (which have traditionally been excluded from the uncertainties reported on these parameters). For uncertainty propagation into the corrected (U-Th)/He date, we use HeCalc (Martin et al., 2023) and assume fully-correlated (r = 1) isotope-specific $F_T$ uncertainties. In Table 4 and the discussion below all uncertainties are reported at 1s. Standard practice in the CU TRaIL over the last several years has been to report 15%
1s uncertainties on eU based on estimates by Baughman et al. (2017). However, how eU uncertainties are reported varies widely across the community and it is common for no uncertainty to be reported on eU data, therefore for comparative purposes, no uncertainty is shown on $eU_{2D}$ in Fig. 8a-c and none is reported in Table E1.

### 6.3.2 Mass and eU

To calculate eU, absolute quantities of U, Th, and Sm must be converted to concentrations using the apatite grain mass, which is computed from V assuming an apatite density (here we use 3.20 g/cm$^3$). Absolute amounts of parent isotope carry an analytical uncertainty, but conventionally the grain mass reported by labs has had no uncertainty attached to it because the geometric uncertainty on V (and therefore on mass) was not well constrained. By applying a correction factor to V based on grain
geometry (0.83 or 0.74) and calculating mass using the corrected V, the mass$_{GCM}$ decreases by the same correction factor as volume. The mass then inherits the same percent uncertainty as volume (20 or 23%, 1s, depending on geometry).

For eU, the smaller mass$_{GCM}$ values (relative to mass$_{2D}$) are translated into larger eU$_{GCM}$ values (relative to eU$_{2D}$). In our example dataset (Table 4), the average eU$_{GCM}$/eU$_{2D}$ is 1.2 for hexagonal grains and 1.4 for ellipsoidal grains. We propagated the analytical uncertainties on the parent isotopes only, as well as the parent isotope and geometric uncertainties, into the eU$_{GCM}$ values. Propagating parent isotope uncertainties only yields average eU uncertainty values of 3% for hexagonal and ellipsoidal grains in this dataset (with a range from 1 to 6%). Including both analytical and geometric uncertainties yields average uncertainties of 15% and 16% for hexagonal and ellipsoidal grains (varying from 14-17%).

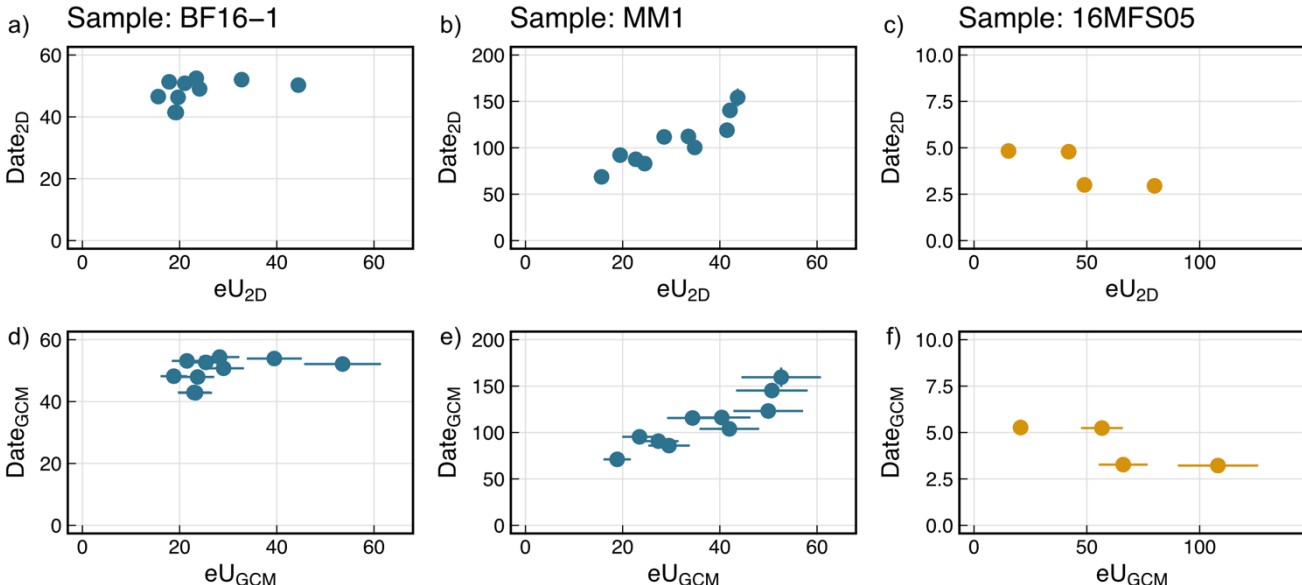

**Figure 8.** Date-eU plots for three samples previously dated in the CU TRaIL showing the effects of corrections and uncertainty estimates on typical AHe data. (a-c) are date$_{2D}$ vs. eU$_{2D}$ plots, while (d-f) are date$_{GCM}$ vs. eU$_{GCM}$ plots. When uncertainty bars are not visible they are on the order of the symbol size, except for the top row where no eU uncertainty is plotted. An idealized hexagonal geometry was used for 2D geometric parameter calculations for the igneous apatite in samples BF16-1 and MM1 (blue circles), while an idealized ellipsoidal geometry was used for the detrital apatite in sample 16MFS05 (yellow circles).

### 6.3.3 Combined F$_T$ values

The combined F$_T$ values are calculated using both the isotope-specific F$_T$ values and the amount of the parent isotopes, because the proportion of the parent isotopes dictates the proportion of the $^4$He atoms that travel different mean stopping distances. The combined F$_T$ values are not used for any additional calculations except R$_{FT}$, but are typically reported in data tables (e.g., Flowers et al., 2022a). For our example dataset, we apply the correction factors in Table 2 based on grain geometry and size to the isotope-specific F$_T$ values, and then use these corrected values to calculate the combined F$_{T,GCM}$ value. F$_{T,GCM}$ is always smaller than F$_{T,2D}$ (F$_{T,GCM}$ / F$_{T,2D}$ = 0.97 and 0.92 for hexagonal and ellipsoidal grains; Table 4).

$F_T$ values have not typically been reported with an uncertainty, because until now the geometric uncertainty on $F_T$ has been poorly quantified. For comparative purposes, we propagated uncertainties into the combined $F_T$ value using the parent isotope uncertainties only, as well as using both parent isotope and geometric uncertainties. For the example dataset, inclusion of analytical uncertainties only yields average uncertainties on the combined $F_T$ of 1% (1s, with a range from 0-3%) for both grain geometries. The propagation of both parent isotope and geometric uncertainties generates average values of 2% for hexagonal grains (varying from 1-3%) and 4% for all ellipsoidal grains (Table 4). Variability in the uncertainties for the combined $F_T$ is due to variability in the total parent isotope uncertainty.

**Table 4. The average percent difference between the 2D and GCM values for example dataset of Table E1.**

| Parameter and Geometry[a] | Avg. GCM/2D[b] | % Analytical uncertainty only[c], 1s | | | % Analytical + Geometric Uncertainty[d], 1s | | | Avg. % uncert. increase[e], 1s |
|---|---|---|---|---|---|---|---|---|
| | | Avg. | Min (%) | Max (%) | Avg. | Min (%) | Max (%) | |
| Mass | | | | | | | | |
| Hex. | 0.83 | NA | NA | NA | 20% | 20% | 20% | NA |
| Ellip. | 0.74 | NA | NA | NA | 23% | 23% | 23% | NA |
| eU | | | | | | | | |
| Hex. | 1.20 | 3% | 1% | 6% | 15% | 14% | 16% | 12% |
| Ellip. | 1.40 | 3% | 2% | 3% | 16% | 16% | 17% | 13% |
| Combined $F_T$ | | | | | | | | |
| Hex. | 0.97 | 1% | 0% | 3% | 2% | 1% | 3% | 1% |
| Ellip. | 0.92 | 1% | 1% | 1% | 4% | 4% | 4% | 3% |
| Corr. Date | | | | | | | | |
| Hex. | 1.04 | 2% | 1% | 6% | 3% | 2% | 7% | 1% |
| Ellip. | 1.09 | 4% | 2% | 6% | 7% | 6% | 8% | 3% |
| $R_{FT}$ | | | | | | | | |
| Hex. | 0.93 | NA | NA | NA | 6% | 6% | 6% | NA |
| Ellip. | 0.85 | NA | NA | NA | 10% | 10% | 10% | NA |

NA indicates "Not Applicable", for example, mass doesn't have any analytical uncertainty on the parent isotopes.

[a] There are N = 20 hexagonal and N = 4 ellipsoidal grains.

[b] The average of the GCM parameter (calculated using the GCM values) divided by the average of the 2D values (calculated using the 2D values) for the example data in Table E1. Values under 1 indicate that the 2D value is larger than the GCM. Values over 1 indicate that the 2D value is smaller than the GCM.

[c] The average of the percent analytical (i.e., parent isotope) uncertainties only for the example data in Table E1.

[d] The average of the percent analytical + geometric uncertainties for the example data in Table E1.

[e] The average percent increase is the difference between the analytical only and analytical + geometric uncertainties.

## 6.3.4 Corrected (U-Th)/He dates

The most rigorous means of calculating $F_T$-corrected (U-Th)/He dates is by incorporating the isotope-specific $F_T$ corrections into the age equation and calculating the corrected date iteratively (Ketcham et

al., 2011). For our example dataset, we used the corrected isotope-specific $F_T$ values (as described above) to calculate the $F_T$-corrected AHe date$_{GCM}$. For the AHe dates, the smaller $F_{T,GCM}$ values (relative to $F_{T,2D}$) are translated into larger corrections for alpha-ejection. Thus, the date$_{GCM}$ values are always older than the date$_{2D}$ values (avg. date$_{GCM}$ / date$_{2D}$ = 1.04 and 1.09 for hexagonal and ellipsoidal grains).


We calculated the uncertainty on the corrected (U-Th)/He dates in two ways for comparative purposes: first by propagating the analytical uncertainties on the parent and daughter only, and next by additionally including the geometric uncertainties on the isotope-specific $F_{T,GCM}$ values and assuming fully-correlated $F_{T,GCM}$ uncertainties (Table 4). For this dataset, we find that propagating only analytical uncertainties yields average uncertainties of 2% and 4% for hexagonal and ellipsoidal grains (varying from 1-6% and 2-6%, respectively). Including both analytical and geometric uncertainties yields average uncertainties of 3% and 7% for the two geometries (with 2-7% and 6-8% variability). The difference in the uncertainty on the date varies so widely because it is dependent on a variety of grain-specific factors—the absolute amounts of U, Th, Sm, and He, as well as grain geometry and size.



### 6.3.5 $R_{FT}$

We applied the correction factors based on grain geometry in Table 2 to $R_{FT}$ values from the example dataset. The $R_{FT,GCM}$ values are always smaller than $R_{FT,2D}$ values ($R_{FT,GCM}$ / $R_{FT,2D}$ = 0.93 and 0.85 for hexagonal and ellipsoidal grains) (Table 4). The uncertainty on $R_{FT}$ is 6% (1s) for hexagonal grains and 10% (1s) for ellipsoidal grains. This parameter is not used in the calculation of (U-Th)/He dates, but the uncertainty should be used during thermal history modeling when possible.


### 6.3.6 Summary

This exercise in which we both 1) correct real AHe data for systematic error associated with the 2D microscopy approach for determining geometric parameters, and 2) propagate geometric uncertainties into the uncertainties on eU and corrected AHe dates reveals a substantial influence of both on some aspects of the results. The most striking outcome is the impact on eU. For example, the eU$_{GCM}$ values of the example dataset increase by 20-40%, resulting in a noticeable shift of data to the right on the date-eU plots (compare Fig. 8a-c with Fig. 8d-f). Moreover, the eU uncertainties when both analytical and geometric uncertainties are included are as much as 17% at 1s, indicating the importance of appropriately reporting and representing eU uncertainties. The influence of systematic error and uncertainties are less substantial for the corrected AHe date than for eU, but are still important. For ellipsoidal grains, the AHe date$_{GCM}$ values are as much as 9% older than the date$_{2D}$ values, with typical uncertainties that increase by as much as 3% when geometric uncertainties are propagated in addition to analytical uncertainties. For hexagonal grains, the corrections and uncertainties are less than for ellipsoidal grains, but non-negligible. Including the geometric uncertainty on the corrected AHe dates may help account for overdispersion in some (U-Th)/He datasets. Properly correcting for systematic error and propagating uncertainties associated with the geometric parameters is an important step for rigorously presenting and interpreting apatite (U-Th)/He data.




## 6.4 The Geometric Correction Method: A practical workflow

The Geometric Correction Method described here and shown in Fig. 9 can be easily integrated into existing (U-Th)/He dating workflows with no additional time, cost, or equipment. This method is most appropriate for grain characteristics like those in this calibration study, with 2D microscopy $F_T$ values > 0.5, length/maximum width ratios of 0.8-3.6 and maximum width/minimum width ratios of 1-1.7. This method also assumes that grain measurements are made in the same manner as this study (Fig. 4) and

that 2D V, $F_T$, and $R_{FT}$ values are calculated using the equations of Ketcham et al. (2011) and Cooperdock et al. (2019). All equations required for the calculations below are in Appendix A. The corrections for systematic error and the uncertainties reported here are only those associated with grain geometry. For $F_T$, additional inaccuracy and uncertainty may be introduced by parent isotope zonation (e.g., Farley et al., 1996), grain abrasion (e.g., Rahl et al., 2003), and grain breakage (e.g., He and

Reiners, 2022), which have potential to be accounted for separately. For mass and the derived eU concentration, additional uncertainty may be associated with the assumed mineral density.

**Step 1. Select grain geometry and GEM category.** Choose apatite grain for analysis. Decide whether the grain is hexagonal or ellipsoidal, which is all that is strictly required to correct the 2D values and

assign uncertainty. However, we strongly encourage assigning a GEM category (Fig. 3) and making other descriptive notes, which can be helpful for data interpretation.

**Step 2. Measure the grain.** Measure the grain using the procedure outlined in Sect. 4.2 and Figure 4.
- Measure the grain length parallel to the c-axis. Only a single length is required; however, if the

grain has an extremely angled or uneven end then two lengths may be measured and their average reported to better capture the average length.
- Measure the apatite grain's maximum width, which is perpendicular to the grain length.
- Note that the grain's maximum width is a factor for selecting the proper $F_{T,GCM}$ uncertainty (see Step 5; Table 2).

**Step 3. Calculate the 2D values.** Calculate 2D microscopy V and isotope-specific $F_T$ values using the hexagonal or ellipsoidal equations of Ketcham et al. (2011) depending on grain geometry. Calculate $R_{FT}$ using the equations of Cooperdock et al. (2019). Note that parent isotope data must first be acquired for the $F_T$ and $R_{FT}$ values to be computed.

**Step 4. Correct the 2D values.** Multiply the 2D microscopy V, isotope-specific $F_T$, and $R_{FT}$ values by the correction factor according to the grain geometry to produce the $V_{GCM}$, $F_{T,GCM}$, and $R_{FT,GCM}$ values (Table 2). Typically, combined $F_T$ values are reported by labs, but the isotope-specific $F_T$ values are required for the most accurate and rigorous calculation of corrected (U-Th)/He dates (Ketcham et al.,

840 2011)

**Step 5. Assign uncertainty**. Attach the uncertainty value to each parameter according to the grain geometry (for $V_{GCM}$, $F_{T,\ GCM}$, $R_{FT,\ GCM}$) and maximum width (for $F_{T,\ GCM}$) (Table 2).

**Step 6. Calculate derived parameters and propagate uncertainties.**

- Calculate mass and eU using the $V_{GCM}$ values. Uncertainty on V should be propagated into the uncertainty on these derived parameters.
- Calculate corrected (U-Th)/He dates using the isotope-specific $F_{T,GCM}$ values. Uncertainty on $F_T$ should be propagated into the final uncertainty on the corrected He date. This uncertainty propagation can be easily accomplished, for example, by using the open access Python program HeCalc for (U-Th)/He data reduction (Martin et al., 2023).

Consider the following example: an apatite grain selected for analysis has a maximum width of 98 μm and a GEM value of B1. The $^{238}F_{T,2D}$ of this grain is 0.67 (see Appendix A and the footnotes of Table E1 for the details of this calculation). The analyst uses Table 2 to select the correction for hexagonal grains (0.97) and performs the following calculation:

$$F_{T,\,GCM} = F_{T,2D} \times \text{correction} = 0.67 \times 0.97 = 0.65$$

The analyst then selects the proper uncertainty from Table 2 based on grain geometry and maximum width. This hexagonal grain is considered medium-sized because it is 98 μm wide, so it has a geometric uncertainty of 3%. The final $^{238}F_{T,\,GCM} = 0.65 \pm 3\%$, if the analytical uncertainty on the absolute amount of $^{238}U$ is not also propagated into the $^{238}F_T$ values. This procedure is repeated for each isotope-specific $F_{T,\,2D}$. The isotope-specific $F_{T,\,GCM}$ values are used in the calculation of the corrected date and both the uncertainty on each isotope-specific $F_T$ and the analytical uncertainty on the parent and daughter isotopes is propagated into the uncertainty on the corrected (U-Th)/He date.

## 7 Conclusions

Uncertainties on the geometric parameters and the data derived from them – V, $F_T$, $R_{FT}$, eU, and corrected (U-Th)/He dates – have not traditionally been included in the reported uncertainties on (U-Th)/He datasets. Nor have such data been corrected for systematic error that might arise from the 2D microscopy approach for determining these values. Although both uncertainties and corrections are important for accurate interpretation of (U-Th)/He datasets, the lack of well-quantified values that can easily be determined and applied to routinely generated data has hindered progress in this area.

In this paper we present the only no-cost, easy-to-implement, and backwards-compatible solution to this problem. The Geometric Correction Method (GCM) is a simple and effective set of corrections and uncertainties derived for V, $F_T$, and $R_{FT}$ values that can be easily incorporated into existing workflows (Fig. 9). This approach corrects these parameters for systematic overestimation and provides an uncertainty that can be propagated into the uncertainty on derived parameters (eU, corrected date). It also can be easily applied to previously published data. These corrections and uncertainties are most appropriate for apatite grains like those in this calibration study, with $F_T > 0.5$, length/maximum width ratios of 0.8-3.6 and maximum width/minimum width ratios of 1-1.7, with grain measurements and parameter calculations performed as in this work.

We also present the Grain Evaluation Matrix (GEM), which is a simple, clear, and consistent tool to systematically characterize apatite grain quality (Fig. 3). Although use of the GEM is not required to apply the Geometric Correction Method, assigning GEM values during grain selection can assist in quickly assessing a sample's overall quality and can help identify potential causes of outlier analyses. The GEM is also an effective teaching tool for those who are new to picking apatite grains, so that the
wide spectrum of possible apatite morphologies is clearly communicated.

The corrections and uncertainties in this study were derived from the regression of 2D and 3D measurements of 237 apatite grains displaying a wide variety of morphologies commonly dated for (U-Th)/He thermochronology. The derived corrections and uncertainties were then applied to a set of real
data analyzed in the CU TRaIL to determine their impact. The primary outcomes are:

1. There is both uncertainty and systematic error associated with the microscopy approach to calculating V, $F_T$, and $R_{FT}$ for apatite.
2. For simplicity, consistency, and efficiency we recommend measuring and using only the apatite
length and maximum width for 2D geometric parameter calculations. For most apatite grains, this method yields lower correction magnitudes and uncertainties than using the length, maximum width, and minimum width measurements because of the underestimation and scatter of 2D minimum width values.
3. Using only the length and maximum width measurements, the true values of V, $F_T$, and $R_{FT}$ for
apatite are all overestimated by the 2D microscopy measurements.
4. All corrections for systematic error and all uncertainties are larger for ellipsoidal grains than for hexagonal grains. For both, V has the largest magnitude of overestimation and uncertainty, followed by $R_{FT}$, and then $F_T$.
5. For a subset of real AHe data (N = 24 analyses), the correction factor for eU typically increases
the eU by ~20% with associated 1s uncertainties of 15-16% when both analytical and geometric uncertainties are included. This has important implications for how data are treated during interpretation and thermal history modeling.
6. For the real dataset, the correction factor for the corrected (U-Th)/He date generally increases the date by 4-9% with associated 1s uncertainties of 3-7% if both analytical and geometric
uncertainties are included. Propagating the geometric uncertainty into the corrected date may help account for overdispersion in some (U-Th)/He datasets.

The geometric corrections and geometric uncertainties are substantial enough that they should be routinely included when reporting eU and corrected (U-Th)/He dates to enhance rigorous data
interpretation. Ongoing work is using this same approach to quantify appropriate corrections and uncertainties for zircon geometric parameters in (U-Th)/He datasets (Baker et al., 2020).

Select apatite for analysis:

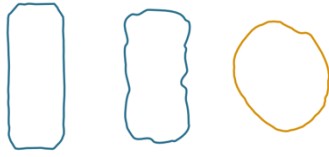

**Step 1.** Select grain geometry and GEM category

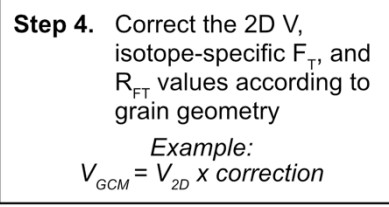

Hexagonal       Ellipsoid

$\downarrow$

**Step 2.** Measure the grain's length and maximum width

$\downarrow$

**Step 3.** Calculate the 2D values

$\downarrow$

**Step 4.** Correct the 2D V, isotope-specific $F_T$, and $R_{FT}$ values according to grain geometry

*Example:*
$V_{GCM} = V_{2D}$ x *correction*

$\downarrow$

**Step 5.** Assign uncertainties to $V_{GCM}$, isotope-specific $F_{T,\ GCM}$, and $R_{FT,\ GCM}$ according to grain geometry (all parameters) and maximum width ($F_{T,\ GCM}$)

*Example:*
$V_{GCM} \pm 1s$ *uncertainty %*

$\downarrow$

**Step 6.** Calculate derived parameters (mass, eU, corrected AHe date) and propagate uncertainties

Figure 9. Flow chart outlining workflow for the Geometric Correction Method.

**Appendix A: Equations required to use the Geometric Correction Method**

All equations necessary to use the corrections and uncertainties are listed below.

Equations for a hexagonal (GEM = A or B) grain from Ketcham et al. (2011), modified to reflect the use of only a maximum width ($W_{max}$; assuming that the minimum width = maximum width) because only a maximum width is used in our preferred Geometric Correction Method, and where we use L to denote grain length instead of H.

$$\Delta V = \frac{1}{6\sqrt{3}}\left(W_{max} - \frac{\sqrt{3}}{2}W_{max}\right)^3 \tag{A1}$$

$$V = LW_{max}\left(W_{max} - \frac{W_{max}}{2\sqrt{3}}\right) - N_p\left(\frac{\sqrt{3}}{8}W_{max}^3 - \Delta V\right) \tag{A2}$$

$$SA = 2L\left(W_{max} + \frac{W_{max}}{\sqrt{3}}\right) + 2W_{max}\left(W_{max} - \frac{W_{max}}{2\sqrt{3}}\right) - N_p\left(\frac{\sqrt{3}}{4}W_{max}^2 + (2 - \sqrt{2})W_{max}^2 + \frac{\sqrt{2}-1}{2\sqrt{3}}W_{max}^2\right) \tag{A3}$$

$$R_{SV} = \frac{3V}{SA} \tag{A4}$$

$$F_T = 1 - \frac{3}{4}\frac{S}{R_{SV}} + \left[(0.2093 - 0.0465N_P)\left(W_{max} + \frac{W_{max}}{\sqrt{3}}\right) + \left(0.1062 + \frac{0.2234S}{S+6(W_{max}\sqrt{3}-W_{max})}\right) \times \left(L - N_P\frac{\frac{W_{max}\sqrt{3}}{2}+W_{max}}{4}\right)\right]\frac{S^2}{V}, \tag{A5}$$

where S is the stopping distance of an alpha particle for a given parent isotope (18.81, 21.80, 22.25, and 5.93 μm for $^{238}$U, $^{235}$U, $^{232}$Th, $^{147}$Sm respectively), $R_{SV}$ is the SV-equivalent spherical radius, and $N_p$ is the number of pyramidal terminations. Equation A5 is used to calculate each isotope-specific $F_T$ value, each with different stopping distance (S).

Equations for an ellipsoidal grain (GEM = C), from Ketcham et al. (2011):

$$V = \frac{4}{3}\pi W_{max}^2 L \tag{A6}$$

$$SA = 4\pi\left(\frac{W_{max}^{2p} + 2W_{max}^p L^p}{3}\right)^{1/p} \text{ with p = 1.6075} \tag{A7}$$

$$R_{SV} = \frac{3V}{SA} \tag{A8}$$

$$F_T = 1 - \frac{3}{4}\frac{S}{R_{SV}} + \left[\frac{1}{16} + 0.1686\left(1 - \frac{W_{max}}{R_{SV}}\right)^2\right]\left(\frac{S}{R_{SV}}\right)^3, \tag{A9}$$

where S is the stopping distance of an alpha particle for a given parent isotope (18.81, 21.80, 22.25, and
5.93 μm for $^{238}$U, $^{235}$U, $^{232}$Th, $^{147}$Sm respectively) and $R_{SV}$ is the SV-equivalent spherical radius.
Equation A9 is used to calculate each isotope-specific $F_T$ value, each with a different stopping distance.

Age equation, from Ketcham et al. (2011):

$$^4He = 8F_{T,238}\,^{238}U(e^{\lambda_{238}t} - 1) + 7F_{T,235}\,^{235}U(e^{\lambda_{235}t} - 1)$$

$$+6F_{T,232}\,^{232}Th\left(e^{\lambda_{232}t} - 1\right) + F_{T,147}\,^{147}Sm\left(e^{\lambda_{238}t} - 1\right) \tag{A10}$$

Equation for combined $F_T$ and $R_{FT}$ from Cooperdock et al. (2019):

$$\frac{S}{R} = 1.681 - 2.428F_T + 1.153F_T^2 - 0.406F_T^3 \tag{A11}$$

$$A_{238} = (1.04 + 0.247[Th/U])^{-1} \tag{A12}$$

$$A_{232} = (1 + 4.21/[Th/U])^{-1} \tag{A13}$$

$$\overline{F_T} = A_{238}F_{T,238} + A_{232}F_{T,232} + (1 - A_{238} - A_{232})F_{T,235} \tag{A14}$$

$$\overline{S} = A_{238}S_{238} + A_{232}S_{232} + (1 - A_{238} - A_{232})S_{235}, \tag{A15}$$

where $S_{238}, S_{232}, S_{235}$ are the weighted mean stopping distances for each decay chain (18.81, 21.80, and
22.25 μm, respectively, for apatite).

$$R_{FT} = \overline{S}/\left(\frac{S}{R}\right) \tag{A16}$$

Equation for eU from Cooperdock et al. (2019):

$$eU = [U] + 0.238[Th] + 0.0012[Sm] \;(or\; 0.0083[^{147}Sm]) \tag{A17}$$

# Appendix B: Additional sample information

**Table B1. Apatite CT scan parameters**

| Mount | 1 | 2 | 3 | 4 | 5 | 6 | 7 | 8 | 9 |
|---|---|---|---|---|---|---|---|---|---|
| Objective | 20X | 20X | 20X | 20X | 20X | 20X | 20X | 20X | 20X |
| Pixel Size (μm) | 0.64 | 0.63 | 0.63 | 0.63 | 0.63 | 0.64 | 0.63 | 0.63 | 0.63 |
| X-Ray Power (W) | 3 | 3 | 3 | 3 | 3 | 3 | 3 | 3 | 3 |
| X-Ray Voltage (kV) | 40 | 40 | 40 | 40 | 40 | 40 | 40 | 40 | 40 |
| Number of Projections | 3201 | 3201 | 3201 | 3201 | 3201 | 3201 | 3201 | 3201 | 3201 |
| Binning | 2 | 2 | 2 | 2 | 2 | 2 | 2 | 2 | 2 |
| Filter | Air | Air | Air | Air | Air | Air | Air | Air | Air |
| Height (pixels) | 1024 | 993 | 993 | 993 | 993 | 993 | 993 | 993 | 993 |
| Width (pixels) | 1024 | 993 | 993 | 993 | 993 | 993 | 993 | 993 | 993 |
| Sample Theta (°) | -180 | -180 | -180 | -180 | -180 | -180 | -180 | -180 | -180 |
| Detector To Sample Distance (mm) | 5.01 | 5.17 | 4.95 | 4.97 | 4.99 | 5.42 | 5.07 | 5.08 | 5.02 |
| Source To Sample Distance (mm) | -4.44 | -4.51 | -4.33 | -4.33 | -4.34 | -4.75 | -4.33 | -4.32 | -4.38 |
| Exposure (s) | 2.1 | 2.5 | 2.3 | 2.3 | 2.0 | 2.7 | 2.3 | 2.5 | 2.5 |
| Total Scan Time (h) | 3.0 | 3.4 | 3.2 | 3.2 | 2.9 | 3.6 | 3.2 | 3.4 | 3.4 |

000

**Figure B1. Grain Evaluation Matrix listing the samples and number of grains for which high-quality CT data (N = 264) were acquired in each category.**

# Geometric Classification

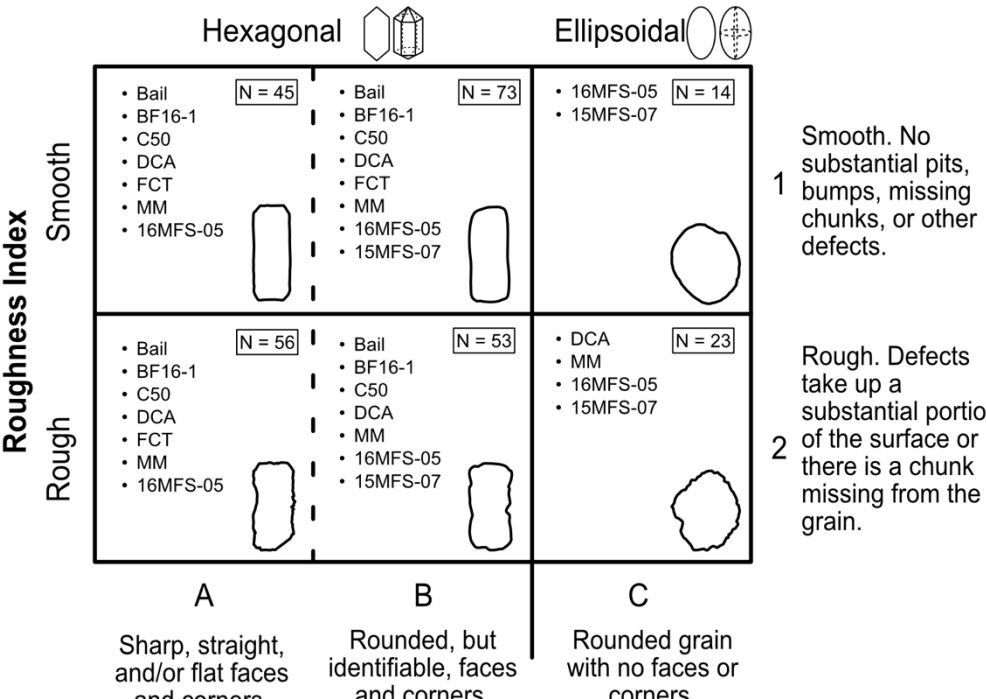

|  | Hexagonal | Ellipsoidal |  |
|---|---|---|---|

**Smooth**

| N = 45 | N = 73 | N = 14 |
|---|---|---|
| • Bail<br>• BF16-1<br>• C50<br>• DCA<br>• FCT<br>• MM<br>• 16MFS-05 | • Bail<br>• BF16-1<br>• C50<br>• DCA<br>• FCT<br>• MM<br>• 16MFS-05<br>• 15MFS-07 | • 16MFS-05<br>• 15MFS-07 |

1 Smooth. No substantial pits, bumps, missing chunks, or other defects.

**Rough**

| N = 56 | N = 53 | N = 23 |
|---|---|---|
| • Bail<br>• BF16-1<br>• C50<br>• DCA<br>• FCT<br>• MM<br>• 16MFS-05 | • Bail<br>• BF16-1<br>• C50<br>• DCA<br>• MM<br>• 16MFS-05<br>• 15MFS-07 | • DCA<br>• MM<br>• 16MFS-05<br>• 15MFS-07 |

2 Rough. Defects take up a substantial portion of the surface or there is a chunk missing from the grain.

**Roughness Index**

| A | B | C |
|---|---|---|
| Sharp, straight, and/or flat faces and corners. | Rounded, but identifiable, faces and corners. | Rounded grain with no faces or corners. |

005

**Appendix C: Regression and uncertainty information for isotope-specific $^{235}F_T$, $^{232}F_T$, and $^{147}F_T$ values**

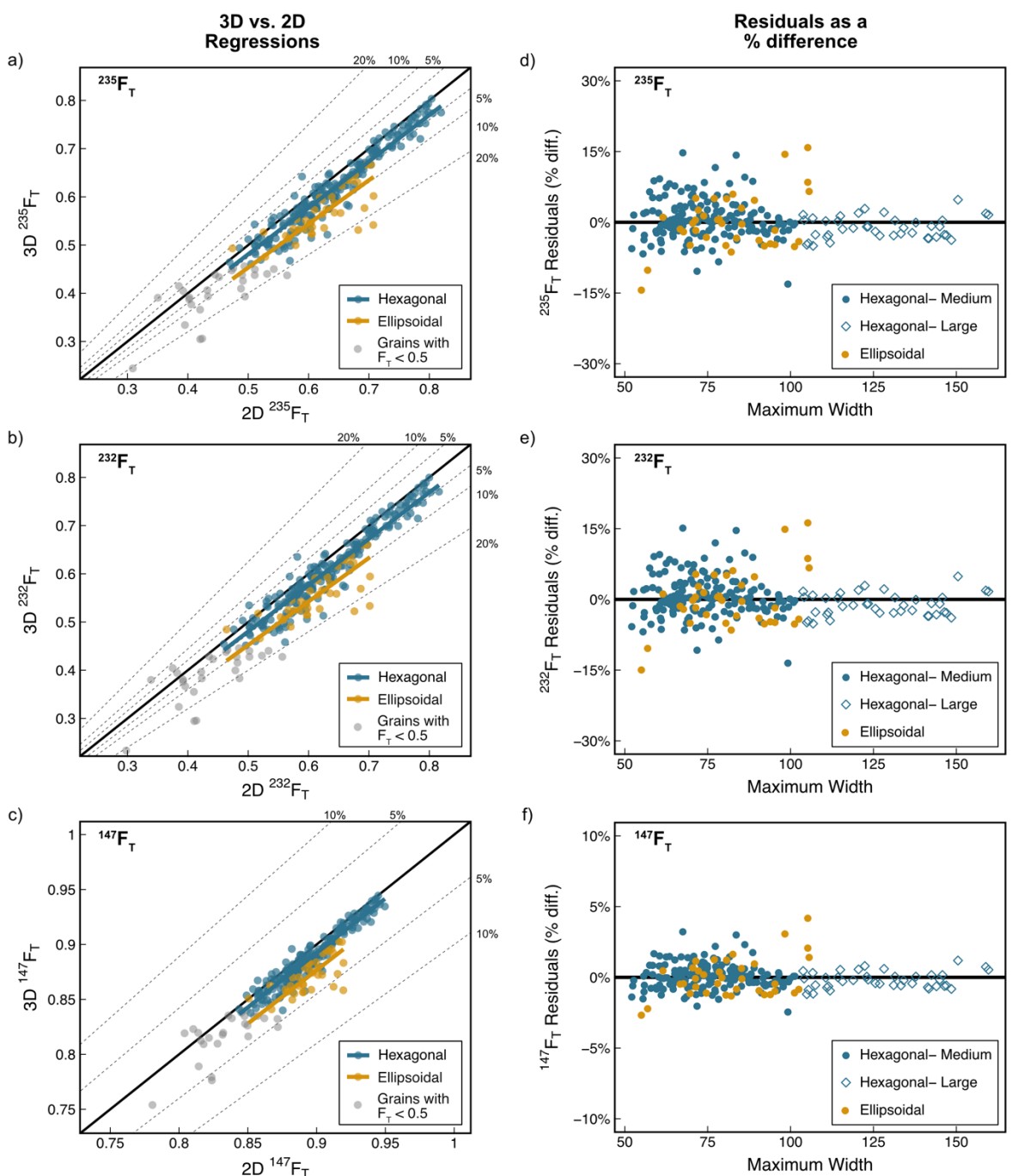

**Figure C1. Plots illustrating how the corrections for systematic error and how uncertainties were determined for each parent isotope-specific $F_T$ (except $^{238}F_T$, which is included in Figure 7). 2D calculations use the maximum width for both width values. Scatter plots of 3D vs. 2D data (N = 264) with regression lines and data distinguished by geometry for (a) $^{235}F_T$ (b) $^{232}F_T$, and (c) $^{147}F_T$. Grains with $F_T < 0.5$ were excluded from the regressions but are included in the plots in light grey. A total of 237 apatite**

**grains are in the regressed dataset. The bold black line is the 1:1 line and the dashed lines mark the percent difference from the 1:1 line. Note that for all regressions, the regression line falls below the 1:1 line, indicating that the 2D-microscopy data overestimate the 3D-CT data. The 2D data can be corrected for systematic error by multiplying the 2D data by the 3D/2D slope. Plots of the difference of each 2D value from the regression line (i.e., the residual) as a percent difference vs. maximum width with data distinguished by geometry and grain size. The bold black line is 0% difference. The standard deviation of the % difference in the residuals of each group is the uncertainty on the parameter.**

**Table C1. Results of Tukey's Highly Significant Difference[a] test to determine if different groups of grains have statistically different slopes.**

| Grouping & Pairs | Difference in Slopes | 95% CI[b] | Adjusted P-value[c] |
|---|---|---|---|
| Volume | | | |
| GEM: Geometric Classification | | | |
| B-A | < 0.001 | [-0.001, 0.001] | 0.922 |
| **C-A** | **0.153** | **[0.153, 0.153]** | **< 0.001** |
| **C-B** | **0.153** | **[0.153, 0.153]** | **< 0.001** |
| Size | | | |
| Medium-Large | 0.011 | [-0.007, 0.029] | 0.213 |
| GEM: Roughness | | | |
| 1-2 | 0.010 | [-0.004, 0.024] | 0.157 |
| $^{238}FT$ | | | |
| GEM: Geometric | | | |
| B-A | < 0.001 | [-.001, 0.001] | 0.922 |
| **C-A** | **0.153** | **[0.153, 0.153]** | **< 0.001** |
| **C-B** | **0.153** | **[0.153, 0.153]** | **< 0.001** |
| Size | | | |
| Medium-Large | 0.011 | [-0.007, 0.029] | 0.231 |
| Roughness | | | |
| 1-2 | 0.010 | [-.001, 0.001] | 0.157 |
| $R_{FT}$ | | | |
| GEM: Geometric | | | |
| B-A | 0 | [-.001, 0.001] | 1 |
| C-A | **0.055** | **[0.055, 0.055]** | **< 0.001** |
| C-B | **0.055** | **[0.055, 0.055]** | **< 0.001** |
| Size | | | |
| Medium-Large | 0.004 | [-.001, 0.001] | 0.213 |
| Roughness | | | |
| 1-2 | 0.004 | [-0.001, 0.009] | 0.157 |

[a] Tukey's Highly Significant Difference tests if slopes are significantly different from each other or not and takes into account the uncertainties on the slopes. Where the null hypothesis, $H_0$, is $\beta_1 = \beta_2$ and the alternative hypothesis, $H_1$, is $\beta_1 \neq \beta_2$.

[b] The 95% confidence interval (CI) of the difference in slopes.

[c] A p-value < 0.05 indicates that $H_0$ can be rejected, i.e., there is a significant difference between the slopes of the pair. If the p-value is > 0.05, this indicates that there is no significant difference between the means of the pair. Bolded pairs of slopes are those with p-values <0.05 and therefore are treated as separate groups.


**Table C2. Uncertainty values (1s) for different groupings of physical variables.**

| Geometry | Size[a] | Roughness | N | Uncertainty |
|---|---|---|---|---|
| **Volume** | | | | |
| **Hex.** | **Medium & Large** | **1 & 2** | **201** | **20%** |
| Hex. | Medium | 1 & 2 | 161 | 20% |
| Hex. | Medium | 1 | 86 | 19% |
| Hex. | Medium | 2 | 75 | 21% |
| Hex. | Large | 1 & 2 | 40 | 23% |
| Hex. | Large | 1 | 18 | 15% |
| Hex. | Large | 2 | 22 | 28% |
| **Ellip.** | **Medium & Large** | **1 & 2** | **36** | **23%** |
| **$^{238}F_T$** | | | | |
| Hex. | Medium & Large | 1 & 2 | 201 | 3% |
| **Hex.** | **Medium** | **1 & 2** | **161** | **3%** |
| Hex. | Medium | 1 | 86 | 3% |
| Hex. | Medium | 2 | 75 | 4% |
| **Hex.** | **Large** | **1 & 2** | **40** | **2%** |
| Hex. | Large | 1 | 18 | 1% |
| Hex. | Large | 2 | 22 | 2% |
| **Ellip.** | **Medium & Large** | **1 & 2** | **36** | **5%** |
| **$R_{FT}$** | | | | |
| **Hex.** | **Medium & Large** | **1 & 2** | **201** | **6%** |
| Hex. | Medium | 1 & 2 | 161 | 6% |
| Hex. | Medium | 1 | 86 | 6% |
| Hex. | Medium | 2 | 75 | 6% |
| Hex. | Large | 1 & 2 | 40 | 7% |
| Hex. | Large | 1 | 18 | 5% |
| Hex. | Large | 2 | 22 | 8% |
| **Ellip.** | **Medium & Large** | **1 & 2** | **36** | **10%** |

[a] Groups in bold are the groups for which uncertainties are reported (i.e., geometry only for V and $R_{FT}$; geometry and grain size for $F_T$).


## Appendix D: In the case of 2D calculations using the minimum and maximum width

We recommend using the maximum width only for apatite 2D calculations for the reasons discussed in Section 6.1. However, for completeness, in this Appendix we present a set of corrections and uncertainties based on our dataset that can be used if both maximum and minimum width measurements are acquired and used to calculate the 2D parameters (Fig. D2, Table D1).


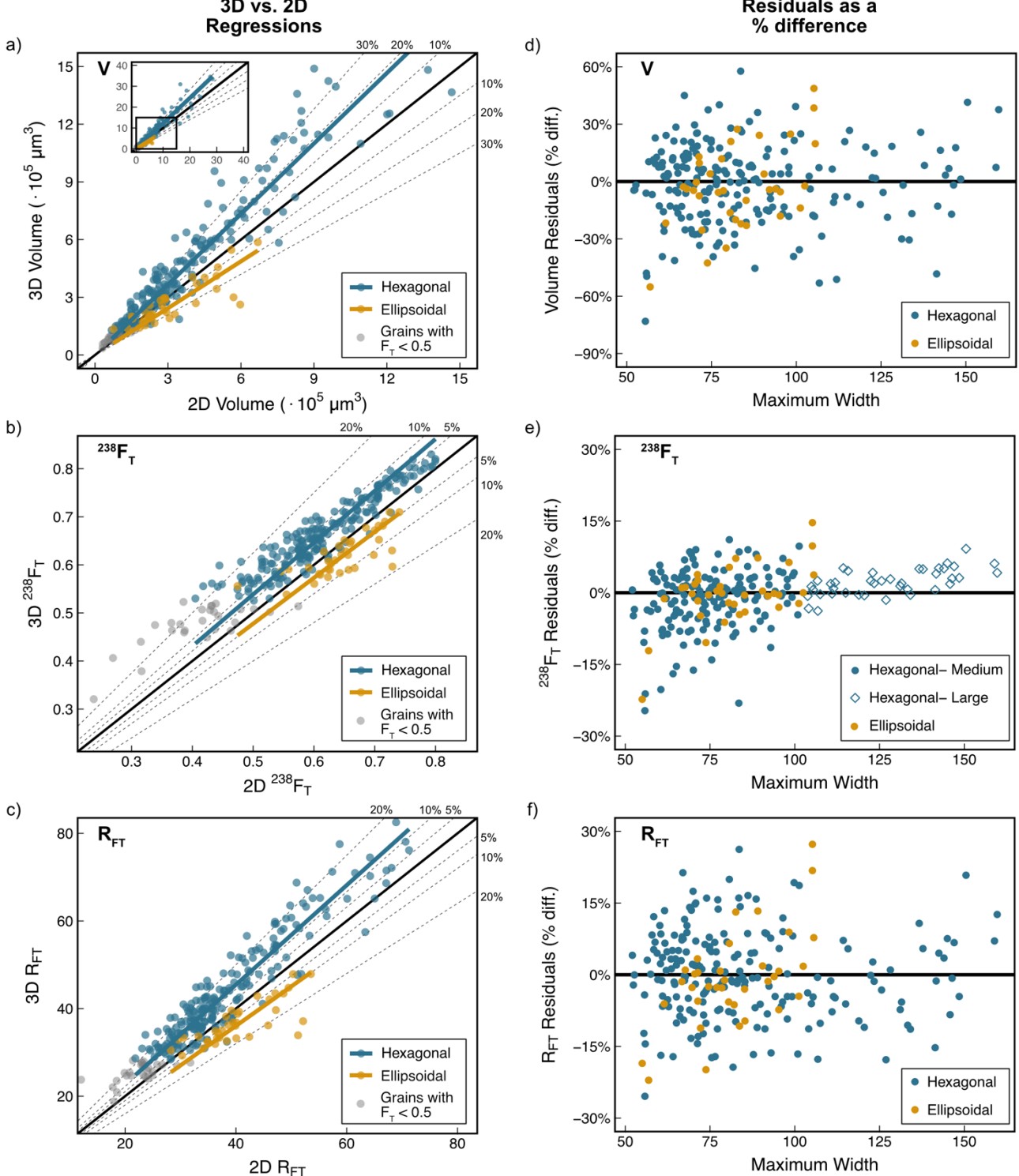

**Figure D1. This figure is the same as Figure 7 except that 2D data were calculated using the length, maximum width, and minimum width values. See Figure 7 and text for additional details.**


**Table D1. Corrections and uncertainties (1s) for all geometric parameters where 2D values are calculated using the length, maximum width, and minimum width.**

## Volume

| Geometry | Correction[a] | % Uncert.[b] (1s) for apatite grains of all sizes |
|---|---|---|
| Volume | | |
| Hex. | 1.27 | 21% |
| Ellip. | 0.86 | 28% |

## Isotope-specific $F_T$ values

| Geometry | Correction | % Uncert. (1s) for medium-sized[c] apatite grains | % Uncert. (1s) for large-sized[d] apatite grains |
|---|---|---|---|
| $^{238}F_T$ | | | |
| Hex. | 1.08 | 6% | 3% |
| Ellip. | 0.96 | 6% | 6% |
| $^{235}F_T$ | | | |
| Hex. | 1.08 | 8% | 4% |
| Ellip. | 0.95 | 7% | 7% |
| $^{232}F_T$ | | | |
| Hex. | 1.08 | 8% | 4% |
| Ellip. | 0.95 | 7% | 7% |
| $^{147}F_T$ | | | |
| Hex. | 1.02 | 2% | 1% |
| Ellip. | 0.98 | 1% | 1% |

## $R_{FT}$

| Geometry | Correction | % Uncert. (1s) for apatite grains of all sizes |
|---|---|---|
| $R_{FT}$ | | |
| Hex. | 1.15 | 9% |
| Ellip. | 0.91 | 10% |


[a] The correction value is the slope of the 3D vs. 2D regression line for each parameter in Figures D1a-c.

[b] The uncertainty is the scatter of the 2D data about each regression line in Fig. D1a-c, calculated as the 1s standard deviation of the % difference of each 2D value from the regression line (Fig. D1d-f).

[c] **"Medium-sized" apatite have maximum widths of 50-100 μm.**

[d] **"Large-sized" apatite have maximum widths of >100 μm.**

## 055  Appendix E: Application of geometric parameter corrections and uncertainties to a real dataset.

**Table E1. Results of applying geometric corrections and uncertainties (1s) to apatite (U-Th)/He data from a suite of samples previously dated in the CU TRaIL.**

| Sample and aliquot [a] | Geo. [b] | Max. Width [c] (µm) | $Mass_{2D}$ [d] (µg) | $Mass_{GCM}$ [e] (µg) | ± [f] (µg) | ± [g] (%) | $eU_{2D}$ [h] (ppm) | $eU_{GCM}$ [i] (ppm) | ±TAU [j] (ppm) | ±TAU [k] (%) | ±TAU + geom [l] (ppm) | ±TAU + geom [m] (%) | $F_{T,2D}$ [n] | $F_{T,GCM}$ [o] | ±TAU [p] | ±TAU [q] (%) | ±TAU + geom [f] | ±TAU + geom [r] (%) | ±TAU ±TAU + geom [s] (%) | $Date_{2D}$ [t] (Ma) | ±TAU [u] (Ma) | ±TAU [v] (%) | $Date_{GCM}$ [w] (Ma) | ±TAU [x] (Ma) | ±TAU [y] (%) | ±TAU + geom [z] (Ma) | ±TAU + geom [aa] (%) | $R_{FT,2D}$ [ab] (µm) | $R_{FT,GCM}$ [ac] (µm) | ± [ad] (µm) | ± [ae] (%) |
|---|---|---|---|---|---|---|---|---|---|---|---|---|---|---|---|---|---|---|---|---|---|---|---|---|---|---|---|---|---|---|---|
| **BF16-1** | | | | | | | | | | | | | | | | | | | | | | | | | | | | | | | |
| a01 | Hex. | 202 | 12.1 | 10.1 | 2.0 | 20% | 19.1 | 23.0 | 0.4 | 2% | 3.3 | 15% | 0.82 | 0.79 | 0.00 | 1% | 0.01 | 1% | 1% | 41.5 | 0.4 | 1% | 42.9 | 0.5 | 1% | 1.0 | 2% | 85 | 79 | 5 | 6% |
| a02 | Hex. | 186 | 10.1 | 8.4 | 1.7 | 20% | 15.6 | 18.8 | 0.4 | 2% | 2.7 | 15% | 0.81 | 0.78 | 0.00 | 1% | 0.01 | 1% | 1% | 46.6 | 0.6 | 1% | 48.2 | 0.6 | 1% | 1.1 | 2% | 79 | 74 | 4 | 6% |
| a03 | Hex. | 188 | 9.9 | 8.3 | 1.7 | 20% | 19.3 | 23.3 | 0.4 | 2% | 3.4 | 15% | 0.82 | 0.79 | 0.00 | 1% | 0.01 | 2% | 2% | 41.4 | 0.5 | 1% | 42.8 | 0.6 | 1% | 1.0 | 2% | 81 | 75 | 5 | 6% |
| a04 | Hex. | 150 | 6.5 | 5.4 | 1.1 | 20% | 19.7 | 23.7 | 0.5 | 2% | 3.4 | 14% | 0.76 | 0.73 | 0.00 | 1% | 0.01 | 1% | 2% | 46.3 | 0.7 | 1% | 48.0 | 0.7 | 1% | 1.2 | 2% | 63 | 58 | 3 | 6% |
| a05 | Hex. | 122 | 3.2 | 2.7 | 0.5 | 20% | 21.1 | 25.4 | 0.6 | 2% | 3.7 | 14% | 0.73 | 0.70 | 0.00 | 1% | 0.01 | 1% | 2% | 50.9 | 0.7 | 1% | 52.7 | 0.7 | 1% | 1.2 | 2% | 54 | 51 | 3 | 6% |
| a06 | Hex. | 189 | 3.2 | 2.7 | 0.5 | 20% | 24.1 | 29.0 | 0.5 | 2% | 4.2 | 14% | 0.71 | 0.69 | 0.00 | 1% | 0.01 | 1% | 1% | 49.1 | 0.6 | 1% | 50.8 | 0.6 | 1% | 1.2 | 2% | 52 | 48 | 3 | 6% |
| a07 | Hex. | 110 | 2.6 | 2.2 | 0.4 | 20% | 23.4 | 28.2 | 0.8 | 3% | 4.1 | 14% | 0.70 | 0.68 | 0.01 | 1% | 0.01 | 2% | 2% | 52.5 | 1.0 | 2% | 54.4 | 1.0 | 2% | 1.5 | 3% | 51 | 47 | 3 | 6% |
| a08 | Hex. | 124 | 2.5 | 2.1 | 0.4 | 20% | 32.8 | 39.5 | 1.0 | 3% | 5.7 | 14% | 0.69 | 0.67 | 0.01 | 1% | 0.01 | 2% | 2% | 52.0 | 1.1 | 2% | 53.9 | 1.1 | 2% | 1.5 | 3% | 49 | 45 | 3 | 6% |
| a09 | Hex. | 106 | 2.2 | 1.8 | 0.4 | 20% | 17.9 | 21.5 | 0.7 | 3% | 3.1 | 15% | 0.67 | 0.65 | 0.01 | 1% | 0.01 | 2% | 2% | 51.3 | 1.2 | 2% | 53.2 | 1.3 | 2% | 1.7 | 3% | 46 | 42 | 3 | 6% |
| a10 | Hex. | 96 | 1.7 | 1.4 | 0.3 | 20% | 44.4 | 53.5 | 1.3 | 3% | 7.9 | 15% | 0.64 | 0.62 | 0.01 | 1% | 0.02 | 2% | 3% | 50.3 | 0.8 | 2% | 52.1 | 0.9 | 2% | 2.0 | 4% | 43 | 40 | 2 | 6% |
| **MM1** | | | | | | | | | | | | | | | | | | | | | | | | | | | | | | | |
| a01 | Hex. | 96 | 5.3 | 4.4 | 0.9 | 20% | 15.7 | 18.9 | 0.4 | 2% | 2.8 | 15% | 0.74 | 0.72 | 0.01 | 1% | 0.02 | 2% | 2% | 68.7 | 1.2 | 2% | 71.1 | 1.3 | 2% | 2.7 | 4% | 58 | 54 | 3 | 6% |
| a02 | Hex. | 168 | 16.0 | 13.3 | 2.7 | 20% | 24.5 | 29.5 | 0.4 | 1% | 4.3 | 14% | 0.84 | 0.81 | 0.00 | 0% | 0.01 | 1% | 1% | 83.0 | 0.8 | 1% | 85.9 | 0.8 | 1% | 1.9 | 2% | 91 | 85 | 5 | 6% |
| a03 | Hex. | 145 | 6.9 | 5.7 | 1.1 | 20% | 41.5 | 50.0 | 0.9 | 2% | 7.2 | 14% | 0.75 | 0.72 | 0.00 | 1% | 0.01 | 1% | 2% | 119.0 | 1.4 | 1% | 123.2 | 1.5 | 1% | 2.8 | 2% | 59 | 55 | 5 | 6% |
| a04 | Hex. | 225 | 20.3 | 16.8 | 3.4 | 20% | 42.1 | 50.7 | 0.7 | 1% | 7.4 | 15% | 0.86 | 0.83 | 0.00 | 1% | 0.01 | 1% | 1% | 140.3 | 1.7 | 1% | 145.2 | 1.8 | 1% | 3.3 | 2% | 105 | 98 | 6 | 6% |
| a05 | Hex. | 107 | 2.8 | 2.3 | 0.5 | 20% | 43.7 | 52.6 | 3.4 | 6% | 8.2 | 16% | 0.71 | 0.68 | 0.02 | 3% | 0.02 | 3% | 3% | 154.2 | 9.6 | 6% | 159.6 | 10.0 | 6% | 10.5 | 7% | 51 | 47 | 3 | 6% |
| a06 | Hex. | 154 | 7.2 | 5.9 | 1.2 | 20% | 22.7 | 27.3 | 1.2 | 5% | 4.2 | 15% | 0.80 | 0.77 | 0.01 | 2% | 0.02 | 2% | 2% | 87.6 | 3.9 | 4% | 90.7 | 4.0 | 4% | 4.4 | 5% | 75 | 70 | 4 | 6% |
| a07 | Hex. | 153 | 7.2 | 6.0 | 1.2 | 20% | 34.8 | 42.0 | 1.6 | 4% | 6.1 | 15% | 0.80 | 0.77 | 0.01 | 1% | 0.02 | 2% | 2% | 100.4 | 3.3 | 3% | 104.0 | 3.4 | 3% | 4.0 | 4% | 74 | 68 | 4 | 6% |
| a08 | Hex. | 154 | 8.6 | 7.2 | 1.4 | 20% | 33.5 | 40.4 | 1.4 | 4% | 5.9 | 15% | 0.81 | 0.78 | 0.01 | 1% | 0.01 | 2% | 2% | 112.2 | 3.3 | 3% | 116.2 | 3.4 | 3% | 4.1 | 4% | 77 | 72 | 4 | 6% |
| a09 | Hex. | 133 | 5.3 | 4.4 | 0.9 | 20% | 28.5 | 34.4 | 1.7 | 5% | 5.2 | 15% | 0.77 | 0.74 | 0.01 | 2% | 0.02 | 2% | 2% | 111.7 | 5.4 | 5% | 115.7 | 5.7 | 5% | 6.1 | 5% | 64 | 60 | 4 | 6% |
| a10 | Hex. | 159 | 10.8 | 9.0 | 1.8 | 20% | 19.5 | 23.5 | 0.9 | 4% | 3.5 | 15% | 0.82 | 0.79 | 0.01 | 1% | 0.02 | 2% | 2% | 92.1 | 3.4 | 4% | 95.4 | 3.5 | 4% | 4.0 | 4% | 82 | 76 | 5 | 6% |
| **16MFS05** | | | | | | | | | | | | | | | | | | | | | | | | | | | | | | | |
| a02 | Ellip. | 74 | 1.5 | 1.1 | 0.3 | 23% | 48.9 | 66.1 | 1.0 | 2% | 10.8 | 16% | 0.66 | 0.60 | 0.00 | 1% | 0.02 | 4% | 4% | 3.0 | 0.1 | 3% | 3.3 | 0.1 | 3% | 0.2 | 6% | 43 | 37 | 4 | 10% |
| a03 | Ellip. | 79 | 1.2 | 0.9 | 0.2 | 23% | 80.0 | 108.1 | 3.2 | 3% | 17.8 | 16% | 0.65 | 0.60 | 0.01 | 1% | 0.02 | 4% | 4% | 3.0 | 0.1 | 3% | 3.2 | 0.08 | 2% | 0.2 | 6% | 43 | 36 | 4 | 10% |
| a04 | Ellip. | 79 | 1.2 | 0.9 | 0.2 | 23% | 15.3 | 20.7 | 0.7 | 3% | 3.4 | 17% | 0.65 | 0.60 | 0.01 | 2% | 0.02 | 4% | 4% | 4.8 | 0.3 | 6% | 5.3 | 0.33 | 6% | 0.4 | 8% | 42 | 36 | 4 | 10% |
| a05 | Ellip. | 91 | 1.3 | 1.0 | 0.2 | 23% | 42.0 | 56.7 | 1.3 | 2% | 9.2 | 16% | 0.66 | 0.61 | 0.00 | 1% | 0.02 | 4% | 4% | 4.8 | 0.1 | 3% | 5.2 | 0.14 | 3% | 0.3 | 6% | 45 | 38 | 4 | 10% |

**All uncertainties reported at the 1s level.**

**All calculations done assuming $F_T$ uncertainties are fully correlated (r = 1).**

**[a] All BF16-1, MM1, and 16MFS05 data are published in Flowers and Kelley (2011), Weisberg et al. (2018), and Collett et al. (2019), respectively.**

[b] Geometry is defined as described in Figure 3 of Ketcham et al. (2011). All GEM A and B grains are hexagonal (hex.) and all GEM C grains are ellipsoidal (ellip.).

[c] Maximum width is measured perpendicular to the length/c-axis.

[d] $Mass_{2D}$ is the mass of the crystal determined by 2D microscopy measurements, the volume assuming the reported grain geometry, and the volume equations and mineral densities in Ketcham et al. (2011).

[e] $Mass_{GCM}$ is computed the same as $mass_{2D}$, but the 2D V is corrected by applying the correction factor in Table 2 based on the grain geometry, and this new volume is used in the mass calculation.

[f] The 1s uncertainty on $mass_{GCM}$ is calculated by propagating the uncertainty on V from Table 2 based on grain geometry through the mass equation.

[g] The 1s percent uncertainty on $mass_{GCM}$.

[h] $eU_{2D}$ is effective Uranium concentration calculated using the $mass_{2D}$. Calculated as U + 0.238*Th + 0.0012*Sm after equation A7 of Cooperdock et al. (2019).

[i] $eU_{GCM}$ is computed the same as $eU_{2D}$ but uses the $mass_{GCM}$ value.

[j] The 1s total analytical uncertainty (TAU, which are the uncertainties on the parent isotopes) on eU. This calculation ignores the negligible contribution from Sm concentration uncertainty and uses 0% geometric uncertainty.

[k] The 1s total analytical percent uncertainty on $eU_{GCM}$.

[l] The 1s TAU + geometric uncertainty on $eU_{GCM}$. This uncertainty includes the total analytical uncertainty and the uncertainty assigned based on grain geometry (Table 2), assumes that the geometric uncertainties on U and Th concentrations are perfectly correlated (r = 1), and ignores the negligible contribution from Sm concentration uncertainty. Although the correlation coefficient will vary with each data set, the dominant contribution to concentration uncertainty comes from the volumetric uncertainty, which is highly correlated. Additionally, assuming perfect correlation yields the maximum possible value, so we use this conservative approach.

[m] The 1s total analytical + geometric percent uncertainty on $eU_{GCM}$.

[n] $F_{T,2D}$ is the combined alpha-ejection correction for the crystal calculated from the 2D parent isotope-specific $F_T$ corrections, the proportion of U and Th contributing to the $^4He$ production, and assuming homogeneous parent isotope distributions using equation A4 in Cooperdock et al. (2019). The parent isotope-specific alpha ejection-corrections were computed assuming the reported grain geometry in this table and the equations and alpha-stopping distances in Ketcham et al. (2011).

[o] $F_{T,GCM}$ is computed the same as $F_{T,2D}$, but uses isotope-specific $F_{T,GCM}$ values corrected by applying the correction factors in Table 2 based on grain geometry and size.

[p] The 1s TAU on $F_{T,GCM}$. This calculation uses 0% geometric uncertainty.

[q] The 1s total analytical percent uncertainty on $F_{T,GCM}$.

[r] The 1s TAU + geometric uncertainty. This uncertainty includes the total analytical uncertainty and uses the parent isotope-specific $F_{T,GCM}$ uncertainties assigned based on grain geometry and size (Table 2).

[s] The 1s total analytical + geometric percent uncertainty on $F_{T,GCM}$.

[t] The corrected (U-Th)/He $date_{2D}$ is calculated iteratively using the absolute values of He, U, Th, Sm, the isotope-specific $F_{T,2D}$ values, and equation 34 in Ketcham et al. (2011) assuming secular equilibrium.

[u] The 1s TAU uncertainty on $date_{2D}$ includes the propagated total analytical uncertainties on the U, Th, Sm and He measurements. Uncertainty propagation done using HeCalc (Martin et al., 2023).

[v] The 1s total analytical percent uncertainty on date$_{2D}$.

[w] The corrected (U-Th)/He date$_{GCM}$ is computed the same as date$_{2D}$, but uses the isotope-specific F$_{T,GCM}$ values corrected by applying the correction factors in Table 2 based on grain geometry and size.

[x] The 1s TAU uncertainty on the corrected (U-Th)/He date$_{GCM}$ includes the propagated total analytical uncertainties on the U, Th, Sm, He measurements. This calculation uses 0% geometric uncertainty. Uncertainty propagation done using HeCalc (Martin et al., 2023).

[y] The 1s total analytical percent uncertainty on the corrected (U-Th)/He date$_{GCM}$.

[z] The 1s total analytical + geometric uncertainty on the corrected (U-Th)/He date$_{GCM}$. This uncertainty includes the propagated total analytical uncertainties on the U, Th, Sm, He measurements and uses the parent isotope-specific F$_{T,GCM}$ uncertainties assigned based on grain geometry and size (Table 2).

[aa] The 1s total analytical + geometric percent uncertainty on the corrected (U-Th)/He date$_{GCM}$.

[ab] R$_{FT,2D}$ is the radius of a sphere with an equivalent alpha-ejection correction as the grain, calculated using the uncorrected parent isotope-specific F$_T$ values in equation A6 in Cooperdock et al. (2019).

[ac] R$_{FT,GCM}$ is computed from R$_{FT,2D}$ by multiplying R$_{FT,2D}$ by the correction factor in Table 2 based on grain geometry.

[ad] The 1s uncertainty on R$_{FT,GCM}$ is assigned based on grain geometry (Table 2).

[ae] The 1s percent uncertainty on R$_{FT,2D}$.

## Code and Data Availability

Raw data and code used to produce the corrections and uncertainties and figures is stored through the Open Science Framework: https://osf.io/fu98s/. All analyses and plots were done in R (Wickham et al., 2019).

## Author Contributions

RMF and JRM conceptualized the project; SDZ curated the data; SDZ and JRM performed the formal data analysis; RMF, JRM, and SDZ acquired funding; SDZ performed the investigation; JRM, SDZ, and RMF developed the methodology; RMF provided supervision; SDZ performed the validation; SDZ did the data visualizations; SDZ and RMF wrote the original draft; RMF, SDZ, and JRM reviewed and edited the manuscript.

## Competing Interests

The authors declare they have no conflict of interest.

## Acknowledgments

We thank Alison Duvall, Lon Abbott, Ray Donelick, and Jacky Baughman for samples. We are grateful to Jennifer Coulombe and Adrian Gestos for their assistance, support, and advice regarding nano-CT and Dragonfly. The use of Blob3D would not have been possible without support from Rich Ketcham and Romy D. Hanna. We appreciate numerous discussions with Morgan Baker. Thanks to Peter Martin for advice and for HeCalc. Rich Ketcham and Christoph Glotzbach provided insightful reviews that
improved the clarity of this manuscript. This work was partially funded by National Science Foundation GRFP DGE-1650115 to Zeigler. Funding for the Zeiss Xradia Versa X-Ray Microscope was provided by NSF CMMI-1726864.

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
