# Peer review of "A practical method for assigning uncertainty and improving the accuracy of alpha-ejection corrections and eU concentrations in apatite (U-Th)/He chronology"

_EGUsphere, 2022_

## Author Response (AR1)

Dear Editor,

Thank you for your prompt decision on our manuscript. For your convivence, here is a list of relevant changes made in the manuscript:

Text changes:

- Added two new references: He and Reiners, 2022; Rahl et al., 2003.
- Added a new section in the results, section 5.1 "Comparison of grain dimensions from 2D microscopy and 3D CT data"
- Edited the discussion section 6.1 to include a discussion of the maximum width vs minimum width and why we chose to use the maximum width only.
- Added a new Appendix section (Appendix D) to present the corrections and uncertainties for 2D calculations that use both widths.
- Included more details regarding how we made our 2D measurements in Sect. 4.2
- Added a brief discussion in section 4.3 regarding uncertainties in the nano-CT measurements.
- Edited numbers throughout the text to reflect the removal of 3 grains that were accidentally included in the grain counts but that were not ever included in the regressions or results.
- Fixed variable naming convention to match the main text and updated explanatory text to define all variables in Appendix A.
- Reordered Appendix such that the "both widths" results are now in Appendix D and the application of the corrections and uncertainties are in Appendix E.

Figure changes:

- Figure 1: completely redone to simply show 3D renders of grains acquired by nano-CT vs. the idealized geometry.
- Figure 2: numbers updated to reflect the removal of 3 grains that were accidentally included.
- Figure 4: added an image of an ellipsoid grain to show how we made our measurements.
- Figure 6: a new figure that shows how 2D grain dimension measurements compared to 3D grain dimension measurements.
- Figure 7, C1, D1: these results figures all have grains < 50µm included in light grey.
- Figure D1: a new results figure which shows the corrections and uncertainties for 2D calculations using both widths (minimum and maximum width)
- Figure B1: numbers updated to reflect the removal of 3 grains that were accidentally included.

Table changes:

- Table 1: added a column with sample names, updated numbers to reflect the removal of 3 grains that were accidentally included.
- Table 3: added summary statistics for 2D vs 3D grain dimension measurements for our study and for Cooperdock et al. (2019).
- Table B1: removed irrelevant rows and edited the units as per Dr. Ketcham's review.
- Table D1: a results table in the same format as the main text Table 2 which shows corrections and uncertainties for 2D calculations using both widths (minimum and maximum width)

Below you will find a point-by-point response to each of the reviewer's comments. Our original response is in bold, the point-by-point response with line numbers reflecting the relevant changes are indented with a dash and the text is in dark blue to differentiate.

Thank you,

Spencer Zeigler

RC1: Rich Ketcham

**We are grateful to Dr. Ketcham for his prompt, thorough, and thoughtful review that will improve our manuscript. We address each of his points and suggestions below, with our responses in bold.**

This is a nice study aimed at creating an easily useable means of getting more reliable numbers for various important parameters for (U-Th)/He thermochronology (volume, FT, equivalent radius, eU) using only traditional optical measurements, rather than going to the trouble of more precise but expensive and/or time-intensive methods. This is certainly within the remit of this journal, and I highly recommend that it be published after addressing the mostly minor concerns below.

Linear relationships between quantities based on 2D measurements and "true" values obtained in 3D are used as conversion factors, and the scatter about those lines is used to characterize the uncertainty. In theory, lab groups can continue doing exactly what they are doing today and use these conversions to get better data and deal with uncertainties more quantitatively.

Accordingly, it's worth asking what other labs might consider barriers to adoption, whether methodological or psychological. Although the latter can't always be fixed, the main concern that comes to mind is that other labs could worry that these conversions may not be appropriate for how they collect their data. Probably the best antidote is providing more and better data. In

particular, some aspects of how measurements are conducted should be better described, particularly for ellipsoidal grains as discussed below.

It would also be good to compare the 2D data to the corresponding 2D-equivalent data extracted from Blob3D (e.g., length corresponding to BoxA, maximum width to BoxB) to verify that there are no errors where there should not be. Some labs may also already be doing 3 measurements (i.e. to include thickness, or w1), and since the authors here also did 3 measurements, I think (based on lines 276-277), and it may be the lack of a third measurement for round grains that gives them a worse result, why not report them? They can also be compared to the 3D BoxC measurement, which will perhaps document that the third measurement is indeed less reliable.

**Thanks for these great suggestions.**

**We'll add comparisons of the Blob3D dimension data to our "2D" dimensions to the tables, figures, and text discussion. Below is a copy of the updated Table 3 with the dimension comparison data in the same style as in Cooperdock (2019) Table 1. We will include scatter plots of the 2D length vs. Box A, 2D maximum width vs. Box B, and 2D minimum width vs. Box C (Reply-Fig1).**

- Added description of how grains were measured on Figure 4 (L510), Section 4.2 (L463-L479).
- Added Figure 6 to show the 3D vs 2D grain dimensions measurements (L755), added language about this to the abstract (L19-26), added the summary statistics to Table 3 (L1130), and added a discussion of the measurements to Section 5.1 (L739).
- Added clarification about our rationale for using the maximum width only in several places, L525-539, L830-840, and L995-1010.
- Added the results using both widths in Appendix D (L1702-1747) and in the main text L827-L838; L926-931).

**Table 3:**

| This Study: 237 apatite grains; CT resolution: 0.64 μm | | | | |
|---|---|---|---|---|
| | avg. | | abs. avg. % | |
| | 3D/2D[b] | 1s | diff.[c] | 1s |
| **All data: 237 grains** | | | | |
| Volume | 0.85 | 0.17 | 19 | 13 |
| $^{238}F_T$ | 0.96 | 0.04 | 4 | 4 |
| $R_{FT}$ | 0.92 | 0.07 | 8 | 6 |
| Length/Box A | 1 | 0.07 | 5 | 6 |
| Width 1/Box B | 0.99 | 0.06 | 5 | 4 |
| Width 2/Box C | 1.09 | 0.14 | 13 | 10 |
| **Hexagonal apatite: 201 grains** | | | | |
| Volume | 0.87 | 0.17 | 18 | 12 |
| $^{238}F_T$ | 0.97 | 0.03 | 4 | 3 |
| $R_{FT}$ | 0.93 | 0.06 | 7 | 5 |
| Length/Box A | 1.01 | 0.07 | 5 | 6 |
| Width 1/Box B | 1 | 0.06 | 4 | 4 |
| Width 2/Box C | 1.11 | 0.12 | 13 | 10 |
| **Ellipsoid apatite: 36 grains** | | | | |
| Volume | 0.75 | 0.17 | 26 | 15 |
| $^{238}F_T$ | 0.92 | 0.05 | 8 | 4 |
| $R_{FT}$ | 0.86 | 0.08 | 15 | 8 |
| Length/Box A | 0.98 | 0.06 | 6 | 3 |
| Width 1/Box B | 0.97 | 0.07 | 6 | 5 |
| Width 2/Box C | 0.97 | 0.16 | 12 | 11 |

**Previous Studies**

| | avg. | | abs. avg. % | |
|---|---|---|---|---|
| | 3D/2D | 1s | diff. | 1s |
| **Cooperdock et al. (2019): 108 apatite grains; CT resolution: 4-5 μm** | | | | |
| Volume | 0.82 | 0.22 | 23 | 16 |
| $^{238}F_T$ | 1.01 | 0.02 | 2 | 2 |
| $R_{FT}$ | 1.02 | 0.07 | 5 | 5 |
| Length/Box A | 0.98 | 0.1 | 4 | 6 |
| Width 1/Box B | 1.03 | 0.07 | 16 | 8 |
| Width 2/Box C | N/A | N/A | N/A | N/A |
| **Glotzbach et al. (2019): 24 apatite grains; CT resolution: 1.2 μm** | | | | |
| Volume | 1.04 | 0.2 | 15 | 13 |
| $^{238}F_T$ | 0.99 | 0.02 | 2 | 2 |
| $R_{SV}$[d] | 0.93 | 0.06 | 8 | 5 |

[Figure]

**Reply-Fig. 1: a) 2D length vs. 3D Box A, b) 2D max. width vs. 3D Box B, and c) 2D min. width vs. 3D Box C.**

[Figure]

As shown in the Reply-Fig. 1c scatterplots, you are correct that the third dimension (Box C) is less reliable than the length and maximum width.

During data analysis for this paper, we had thoroughly explored the use of both widths for the corrections and uncertainties, but decided against it for the reasons discussed below. Thanks for pointing out that the rationale for this decision should be explained in the paper.

We will revise the paper to include corrections and uncertainties for measurements that use all three dimensions (i.e., a length, a maximum width, and a minimum width) for those labs that would like to continue measuring all 3 dimensions (see Reply-Table 1 and Reply-Figure 2 below). The magnitude of both the correction and the uncertainties are larger for hexagonal grains when using all 3 dimensions (Reply-Table 1). For ellipsoid grains, the magnitude of the correction is slightly smaller, but the uncertainties are larger or the same when using all 3 dimensions. Measuring the maximum width is much more consistent and "true" than measurements of the minimum width (Reply-Fig1c). Due to the ease and reliability of the measurements of the maximum width and because the corrections and uncertainties do not suffer from this simplifying assumption we will continue recommending using the maximum width approach. However, we will include further discussion and additions to Figure 4 in the manuscript to make it clearer how we measured the second width and will include the corrections and uncertainties that use all three dimensions in the Appendix.

**Reply-Table 1:**

Corrections and uncertainties for 2D values calculated using the minimum and maximum width.

**Volume**

| Geometry | Correction[a] | % Uncert.[b] (1s) for apatite grains of all sizes |
|---|---|---|
| Volume | | |
| Hex. | 1.27 | 21% |
| Ellip. | 0.86 | 28% |

**Isotope-specific $F_T$ values**

| Geometry | Correction | % Uncert. (1s) for medium-sized[c] apatite grains | % Uncert. (1s) for large-sized[d] apatite grains |
|---|---|---|---|
| $^{238}F_T$ | | | |
| Hex. | 1.08 | 6% | 3% |
| Ellip. | 0.96 | 6% | 6% |
| $^{235}F_T$ | | | |
| Hex. | 1.08 | 8% | 4% |
| Ellip. | 0.95 | 7% | 7% |
| $^{232}F_T$ | | | |
| Hex. | 1.08 | 8% | 4% |
| Ellip. | 0.95 | 7% | 7% |
| $^{147}F_T$ | | | |
| Hex. | 1.02 | 2% | 1% |
| Ellip. | 0.98 | 1% | 1% |

**$R_{FT}$**

| Geometry | Correction | % Uncert. (1s) for apatite grains of all sizes |
|---|---|---|
| $R_{FT}$ | | |
| Hex. | 1.15 | 9% |
| Ellip. | 0.91 | 10% |

**Reply-Fig. 2: 2D calculations using both the maximum and minimum widths.**

[Figure]

A particular concern is how this method works for ellipsoidal grains. The residuals for hexagonal crystals are fairly evenly scattered as a function of size, but the residuals for ellipsoids are clearly

structured, with 2D underestimating volume, FT, and RFT for smaller (less-wide) grains and over-estimating for larger ones. This is partly traceable to the residual lines being forced to pass through the origin, and probably reflects a transition in the characteristics of the error. It's hard to gauge exactly because it's not 100% clear how the microscope measurements are done, and in particular what assumption is made for the third dimension. For example, if the third radius is assumed to be equal to the second, maybe this is a more reasonable assumption for small grains than large ones; perhaps a large flat grain is more likely to be accepted for analysis than a small flat one, as the larger dimensions of the latter are already at the edge of useability. (If this is the case, maybe a different estimate of the third radius would give a better result). Anyway, comparing the 3D-measured third radius to whatever the authors are using might help isolate error components for ellipsoidal grains (e.g. flatness vs. angularity), and point to a way for reducing them.

**Thank you for the ideas on how to identify the source of error for ellipsoid grains. The structure in the residuals that you note still exist despite the use of the third dimension (see Reply-Fig. 2). This point deserves more investigation in the future, and we will try to mention it in the discussion**

**Interestingly, as shown in Reply-Fig. 1c, the minimum width vs. Box C plots do show that our method of measuring the third dimension causes a systematic overestimation of that dimension for ellipsoid grains. This overestimation probably has to do with how exactly the second width is measured, so we will include extra description and additions to Figure 4 to make this clearer.**

- Clarified our assumption for the third width in the main text (L528, L814) and in Appendix A (L1480)
- Compared our third measured dimension to the 3D dimensions in Figure 6 (L755) and in the main text Discussion (L739).
- We were unable to identify the error components for ellipsoid grains.

It would also be worth looking into and discussing briefly how the approach in this paper meshes with the recently published work by He and Reiners in this journal on FT estimates for broken grains.

**Agreed. We will add some discussion about this.**

- Added citation on L1344 and L229.

[Figure 1] This figure has several components that are difficult to understand. What is being (schematically) measured, and how is that applied to the ideal model? For the hexagonal case, on the left there is the appearance of using a bipyramidal model on a grain without pyramidal terminations, but a component of error for that is not shown (wouldn't it underestimate volume if

L is modeled as point-to-point rather than face-to-face?), and on the right, the "error" component consists of a bad interfacial angle (but that's not how crystal faces work) and a mismeasured w2 (unless you're assuming w2 is assumed rather than measured?). For the ellipsoid, the L measurement does not appear to correspond to the true long axis; if one rotates the L and w1 measurements 45 degrees counter-clockwise the ellipsoid would fit better. Was this intentional? It seems to confuse the issue of where one should measure very irregular to ellipsoidal grains. Lastly, the text "top-down view" is confusing; maybe "along c axis view"?

**We appreciate your detailed comments concerning this figure. This figure was intended to be a schematic representation of the issue at hand—microscopy measurements do a poor job accurately representing the true volume and/or surface area of a grain, but it is clear that we did not accomplish this goal. We will remove the pyramidal terminations from the 'idealized geometry' on the hexagonal case and rethink how to represent the ellipsoid grains. We will focus our efforts on how to represent the conceptual issue while remaining accurate to grain geometries and corresponding 2D measurements.**

- We completely redid Figure 1 (L130) with your suggestions in mind. Since the goal of this figure was to communicate that microscopy measurements do a poor job accurately representing the true volume and/or surface area of a grain we decided to simplify and compare the 3D models from CT to the idealized geometry. We kept the doubly terminated idealized hexagonal geometry to communicate that assigning terminations is often complicated and varies between individuals who are doing the measurements.

[line 146-7] It's more common to reference commercial software by company name, and perhaps version number; e.g., Dragonfly (Object Research Systems, v2022.2). But maybe this journal has a different convention?

**Thank you for the formatting suggestion, we will use it as we are unaware of any specific convention from this journal.**

- Changed citation format on L251, L579.

[line 197-214; Figure 4] It's awkward to utilize and discuss the maximum width before you have defined it, and how it's measured. Maybe refer back to Fig 1 to define it (w1) and forward to section 4.2 for how it's measured?

**Excellent suggestion, we will be sure to define the maximum width before we use it.**

- Defined maximum width in Figure 1 (L130) and in the main text (L334).

[line 274-275] For angular ellipsoid grains, what is measured as the long axis? For example, for the center grains in cell C1 of Fig 3b, is the long axis the corner-to-corner distance, or perpendicular to what appears to be a width? How about for the left-column crystals in cell C2?

Might different conventions for measuring such crystals lead to different 2D-to-3D conversion factors?

**We will clarify how the measurements were made in the paper. Generally, a "width" is identified first, which would theoretically be parallel to any flatly broken surface, since apatite grains tend to break perpendicular to the c-axis. The long axis is then identified as the axis perpendicular to what appears to be a width. Of course, as you point out, identifying a width can be extremely difficult (as shown by those left column grains in the manuscript's Figure 3b, cell C2), so we can expect some variability in the 2D-to-3D conversion factors to arise from this issue. We tried to capture this variability in our dataset, which is why grains like this were not removed as long as the three workers who each measured these grains all had methods of measuring these grains that seemed legitimate and sensical.**

- Discussed the difficulty of measuring ellipsoid grains and generally clarified our measurement process in Section 4.2 (L461-479) and in Figure 4 (L509).

[line 294-304] What did you use for the third dimension for ellipsoidal grains? Did you use the 3rd dimension under the microscope, or did you assume that third radius was equal to the second?

**We measured a 3rd dimension for all ellipsoid grains under the microscope and will include this data in the raw data sheet available through OSF and in the summary tables (shown in the updated Table 3). For the 2D calculations presented in the main text, we assumed that the third radius was equal to the second radius (i.e., the "maximum width") for the reasons discussed above. We will make this clearer in the main text.**

- Clarified our assumption for the third width in the main text (L528, L814) and in Appendix A (L1480).

[line 321 (and Appendix B)] Is there a reason such a low X-ray energy was selected (40 kV)? Using non-filtered, low-kV X-rays may have worsened the artifacts. Also, in the Appendix table, are Height and Width really in μm? Those dimensions look more like pixels. Additionally, the X, Y and Z positions are not innately interesting or informative (they are where the stage was positioned for the scan), and could be safely omitted.

**Low X-Ray energy in combination with very small source-sample-detector distances was determined to provide the highest resolution data at the lowest cost. Thank you for noting the units, those will be fixed and the unnecessary rows removed.**

- We updated Table B1 to include these edits (L1621).

[line 349-351] Sorry about the wire; maybe next time try aluminum. Although the text mentions extensive culling of bad data due to artifacts, are there still uncertainties in the CT measurements?

**We discussed this possibility with the CT technicians at the MIMIC Facility at CU Boulder who helped us run our samples. They told us that for preventative maintenance to pass, they need to achieve < 0.5% difference between the actual and measured grid size. Over time, this can drift but generally stays at < 1% difference. We then assumed a ± 1% uncertainty in our CT measurements and performed simulations in Blob3D by varying the voxel size similar to those done in Cooperdock et al. (2019). We found that any uncertainties present in the CT data translate to negligible differences in the relevant values output by Blob3D and are vanishingly small compared to the uncertainties in the 2D measurements.**

- Added a discussion of uncertainties in CT measurements in Section 4.3 (L596-603).

[line 365] Add "grains" to end of sentence.

**We will add this.**

- Done, L568.

[line 403-407] It would be good to also report how well the 2D measurements correspond to the caliper dimensions reported in Blob3D (e.g. Cooperdock et al. 2019, Table 1).

**Excellent suggestion, these values are now included in our updated Table 3 attached above.**

- These values are now included in Figure 6 (L753), Table 3 (L1131).

[line 693] Replace ", however," with ", but". Otherwise, it's a run-on sentence.

**We will correct this.**

- Replaced the comma with a semicolon to correct the run on (L1355).

RC2: Christoph Glotzbach

**We appreciate Dr. Glotzbach's time spent writing this prompt, constructive, and thorough review that will help us improve our paper. We have replied to each of his suggestions in bold below. Figures and tables are included inline and as a higher resolution PDF.**

Dear Authors,

Overall, I found the manuscript well-written, and scientifically interesting and the data is of high quality. I still have some comments and suggestions that I would like to see addressed by the authors:

- I do not get why small grains (<50 μm) are initially considered and later on the results not included. Either include them in the interpretation or (my favorite) do not include them in the manuscript since as you said such grains are commonly not used for apatite (U-Th)/He dating.

**We initially included grains < 50 μm for our analytical work to evaluate if 2D measurements deviated from 3D measurements significantly more at small grain sizes, thus further justifying not analyzing grains that small. We found that they are excessively scattered and therefore we do not include these grains in the regressions to avoid biasing the results just to accommodate grains we don't suggest analyzing. We will explain this more fully in the text and will consider whether/how to include these data in the figures and tables.**

- We included the small grains on all results plots in light grey and clarifying text in the associated captions (Figure 7 (L945-955); Figure C1 (L 1640-1650); Figure D1 (L1730)). We also updated the text to explain this decision more carefully (L561-567).

- The manuscript provides a huge amount of CT data at a very high resolution, and I like to encourage the authors to do further analysis and explore the reasons for their overestimation and statistics of grain geometries.

**As per the review left by Dr. Ketcham, we will include further analysis of the grain dimensions in 2D vs. 3D (see the updated Table 3 and Reply-Figure 1 attached below). Changes to the manuscript's Figure 1 will focus on showing the reasons for overestimation more precisely and we will include some discussion about the reasons for overestimation in the text.**

- Added description of how grains were measured on Figure 4 (L510), Section 4.2 (L461).
- Added Figure 6 to show the 3D vs 2D measurements (L755), added language about this to the abstract (L19-26), added the values to Table 3 (L1130), and added a discussion of the measurements to Section 5.1 (L739-750)

- Added clarification about our rationale for using the maximum width only in several places, L525-539, L830-840, L995-1010.
- Added the results using both widths in Appendix D (L1702-1747) and in the main text L926-931.

**Table 3:**

This Study: 237 apatite grains; CT resolution: 0.64 μm

| | avg. 3D/2D[b] | 1s | abs. avg. % diff.[c] | 1s |
|---|---|---|---|---|
| **All data: 237 grains** | | | | |
| Volume | 0.85 | 0.17 | 19 | 13 |
| $^{238}F_T$ | 0.96 | 0.04 | 4 | 4 |
| $R_{FT}$ | 0.92 | 0.07 | 8 | 6 |
| Length/Box A | 1 | 0.07 | 5 | 6 |
| Width 1/Box B | 0.99 | 0.06 | 5 | 4 |
| Width 2/Box C | 1.09 | 0.14 | 13 | 10 |
| **Hexagonal apatite: 201 grains** | | | | |
| Volume | 0.87 | 0.17 | 18 | 12 |
| $^{238}F_T$ | 0.97 | 0.03 | 4 | 3 |
| $R_{FT}$ | 0.93 | 0.06 | 7 | 5 |
| Length/Box A | 1.01 | 0.07 | 5 | 6 |
| Width 1/Box B | 1 | 0.06 | 4 | 4 |
| Width 2/Box C | 1.11 | 0.12 | 13 | 10 |
| **Ellipsoid apatite: 36 grains** | | | | |
| Volume | 0.75 | 0.17 | 26 | 15 |
| $^{238}F_T$ | 0.92 | 0.05 | 8 | 4 |
| $R_{FT}$ | 0.86 | 0.08 | 15 | 8 |
| Length/Box A | 0.98 | 0.06 | 6 | 3 |
| Width 1/Box B | 0.97 | 0.07 | 6 | 5 |
| Width 2/Box C | 0.97 | 0.16 | 12 | 11 |

Previous Studies

| | avg. 3D/2D | 1s | abs. avg. % diff. | 1s |
|---|---|---|---|---|
| **Cooperdock et al. (2019): 108 apatite grains; CT resolution: 4-5 μm** | | | | |
| Volume | 0.82 | 0.22 | 23 | 16 |
| $^{238}F_T$ | 1.01 | 0.02 | 2 | 2 |
| $R_{FT}$ | 1.02 | 0.07 | 5 | 5 |
| Length/Box A | 0.98 | 0.1 | 4 | 6 |
| Width 1/Box B | 1.03 | 0.07 | 16 | 8 |
| Width 2/Box C | N/A | N/A | N/A | N/A |
| **Glotzbach et al. (2019): 24 apatite grains; CT resolution: 1.2 μm** | | | | |
| Volume | 1.04 | 0.2 | 15 | 13 |
| $^{238}F_T$ | 0.99 | 0.02 | 2 | 2 |
| $R_{sv}$[d] | 0.93 | 0.06 | 8 | 5 |

**Reply-Fig. 1: a) 2D length vs. 3D Box A, b) 2D max. width vs. 3D Box B, and c) 2D min. width vs. 3D Box C.**

[Figure]

- I do miss a discussion about the reasons this study found an overestimation in volume, FT, and radius. The authors did only consider the maximum width and since grains are usually not symmetric perpendicular to the c-axis, their finding is not a surprise. They should have noticed this since they have very high-resolution CT data, which they could use to e.g. make some statistics from the extracted 3D volumetric data (comparing W1 and W2 of hexagonal grains and W1 and L of ellipsoid grains).

**Thanks for this suggestion. As also noted in the response to review 1, we will include a more detailed explanation of why we decided to use the "maximum width only" 2D measurements and how this decision impacts the corrections and uncertainties. Additionally, we will include statistics and plots comparing our 2D measured grain dimensions to the 3D measured grain dimensions that are output by Blob3D which can be found attached to this reply in the updated Table 3 and Reply-Figure 1 above.**

- Added description of how grains were measured on Figure 4 (L510), Section 4.2 (L461).
- Added Figure 6 to show the 3D vs 2D measurements (L755), added language about this to the abstract (L19-26) added the values to Table 3 (L1130), and added a discussion of the measurements to Section 5.1 (L739).
- Added clarification about our rationale for using the maximum width only in several places, L525-539, L830-840, L995-1010.
- Added the results using both widths in Appendix D (L1702-1747) and in the main text L926-931.

- Anyway, a simple additional measurement of the minimum width could correct the data and therefore no correction is required. In my study, I measured the minimum and maximum width and therefore could not find a systematic overestimation. In case you have measured the minimum width (at least you stated this in your manuscript) please provide all geometric estimates based on the minimum and combined minimum and

maximum (volume, FT, and sphere-equivalent radius) measurements and a comparison with the CT data.

**We thank you for noting this clear gap in our work. You will find the resulting corrections and uncertainties calculated using all three dimensions (L, W1, W2) attached to this reply in Reply-Table 1 and Reply-Fig. 2. Using all three dimensions does not negate the need for a correction, it simply changes the direction and magnitude. When we calculate the 2D volume, FT, and RFT using all three dimensions for hexagonal grains, we see that 2D measurements underestimate the 3D parameters. This results in a larger correction and larger uncertainties. However, for ellipsoid grains, using all three dimensions results in a smaller magnitude overestimation, while the uncertainties are larger or stay the same. We will include these results in the manuscript and include a more detailed description of why we favor using the maximum width only.**

- We added the results using all three dimensions to Appendix D (L1705-1750). Additionally, we included a brief discussion of these results and why we favor the maximum width only approach on L740-650, L830-841, L926-931, and L996-1011.

**Reply-Table 1:**

Corrections and uncertainties for 2D values calculated using the minimum and maximum width.

**Volume**

| Geometry | Correction[a] | % Uncert.[b] (1s) for apatite grains of all sizes |
|---|---|---|
| Volume | | |
| Hex. | 1.27 | 21% |
| Ellip. | 0.86 | 28% |

**Isotope-specific $F_T$ values**

| Geometry | Correction | % Uncert. (1s) for medium-sized[c] apatite grains | % Uncert. (1s) for large-sized[d] apatite grains |
|---|---|---|---|
| $^{238}F_T$ | | | |
| Hex. | 1.08 | 6% | 3% |
| Ellip. | 0.96 | 6% | 6% |
| $^{235}F_T$ | | | |
| Hex. | 1.08 | 8% | 4% |
| Ellip. | 0.95 | 7% | 7% |
| $^{232}F_T$ | | | |
| Hex. | 1.08 | 8% | 4% |
| Ellip. | 0.95 | 7% | 7% |
| $^{147}F_T$ | | | |
| Hex. | 1.02 | 2% | 1% |
| Ellip. | 0.98 | 1% | 1% |

**$R_{FT}$**

| Geometry | Correction | % Uncert. (1s) for apatite grains of all sizes |
|---|---|---|
| $R_{FT}$ | | |
| Hex. | 1.15 | 9% |
| Ellip. | 0.91 | 10% |

**Reply-Fig. 2: 2D calculations using both the maximum and minimum widths.**

[Figure]

- The authors stated, that the provided corrections might be simply transferable to other labs/users. I doubt that this is that easy and only possible in case the protocol is similar. Since the authors have not at least provided information on the protocols of other labs, this is just a guess. I personally do not like that you suggest a protocol in which only the maximum width of grains is measured and afterward the overestimation is corrected. This does introduce scatter to the data. Instead, I would suggest measuring at least both the minimum and maximum width, which should remove the overestimation. Those that want to be even more exact can measure according to the 3D-He approach or even use CT scanning.

**As noted above, we will include more details on our measurement procedure and add the data and discussion regarding the use of both widths to the text. Thanks for pointing out the need for additional information here.**

- As noted above, we included a discussion of how we performed our measurements, the results for using a minimum and maximum width, a discussion of those results, and our justification for using the maximum width only in several places throughout the manuscript (line numbers for changes in point-by-point responses above).

- It would be consequent and useful to show the consequence of your applied corrections for the cooling history. I would therefore suggest that you model cooling histories from the presented data before and after the corrections are done.

**The magnitude of the effect of our corrections and uncertainties on thermal history modeling depends strongly on how the uncertainties are input into the thermal modeling program and the cooling history experienced by the sample. There is a lot of variability in how modeling is done across the community, so we would prefer to leave this potentially complicated exercise for future users of the method. We will be exploring this in the future as we routinely incorporate this method in our own data reduction and thermal history modeling procedures.**

I hope you find my comments and suggestions helpful.

**Yes, thank you.**

Technical corrections:

Line 52: Please make it more clear why you preferentially overestimate the grain size. My guess is that the grain will lay down with the c-b axis parallel to a horizontal. This can be easily corrected if you would use low-adhesive tape and rotate around the c-axis the measure also the a-axis.

**We will add text to clarify this point. Additionally, we will update Figure 4 and the text to explain how we measured the second width.**

- We added text to Section 4.2 (L460-478) to discuss how we performed our measurements. Figure 1 (129) and Figure 4 (L510) are also updated to convey more information about our measurement procedure.

Anyway, explain in Fig. 1, how you get to the red 'overestimated area'.

**We will rework Figure 1 based on your and Dr. Ketcham's reviews.**

- We completely redid Figure 1 (L130) with your and Dr. Ketcham's suggestions in mind. Since the goal of this figure was to communicate that microscopy measurements do a poor job accurately representing the true volume and/or surface area of a grain we decided to simplify and compare the 3D models from CT to the idealized geometry.

Line 131: I would say that according to present knowledge radiation damage does have a large effect in case of slow cooling or long stay in the PRZ.

- Edited this line on L233.

Line 175: It is not clear at this point why it is important to analyze apatites from different lithologies and ages? Please clarify.

**We chose a wide range of lithologies and ages to ensure we were selecting apatites with grain characteristics that are representative of all the types of apatites dated regularly with (U-Th)/He thermochronology. Our lab's research tends to focus on older basement samples, and we wanted to include igneous apatite from young igneous systems as well as detrital apatite to ensure that we weren't inadvertently biasing the outcomes of this work. We will clarify this point during revisions.**

- We added clarification to this point on L283.

Line 223: Maybe better say 'code' instead of 'single value'.

**Will change.**

- Changed this on L394.

Line 254-255: I do see the benefit of having such a classification scheme, but it is not only useful for newcomers. Just say that this simple approach might be useful to easily report the overall morphological quality of grains in studies reporting apatite (U-Th)/He data.

**Good suggestion. We will add this language, but do prefer to also retain mention of the particular value to newcomers because learning to pick apatite is difficult, partially because of the wide range of morphologies apatite can exhibit, which the GEM can help with.**

- Updated this language on L427-429.

Line 267: That is a very important point, the reported corrections are only applicable under these conditions. Therefore, I am not sure if your method is that 'simple and practical and even applicable retroactively. Please make this clear in the abstract and elsewhere.

**We will clarify. However, to our knowledge, we use the same approach that most labs do for determining grain length and width measurements.**

- We made clarifying changes to this point on L24, L456-460, L1090-1092, L1323-1340, and L1416-1419.

Line 277: I agree that this is difficult on a glass slide only, but is quite simple with low adhesive double-sided tape. The extra work on doing this is at a maximum 1-2 minutes, nothing really difficult, and neglectable compared to the time needed to search and pick suitable grains. I also do not see why you are only using 'maximum width', which consequently results in an overestimation of volume, FT and radius.

**We will clarify our decision to use the maximum width only as discussed above. We will also discuss and illustrate the difficulty of measuring the second width well (updated Table 3, Reply-Figure 1c).**

- We clarified our decision to use the maximum width only in several places in the main text (L467-479, L741-750, L997-1011) and through the inclusions of Figure 6 (L754) and Table 3 (L1131).

Line 351: I would suggest reporting a few of the reconstructed apatites.

**We don't feel comfortable reporting these data that are unreliable due to acquisition with a flawed analytical procedure.**

Line 361: This procedure is not clear to me, you have selected grains <50 μm in width and analyzed them and finally not considering them. Does not make sense to me. It would make sense to say from the start that you have not included grains <50 μm.

**We will more clearly explain why we initially included grains <50 μm as noted above, and will consider including these data in some form.**

- We included the small grains on all results plots in light grey and clarifying text in the associated captions (Figure 7 (L945-955); Figure C1 (L 1640-1650); Figure D1 (L1730)). We updated the text to explain this decision more carefully (L561-567).

Line 364: Change to 'typically'.

**We will reword.**

- Reworded this section (L662-669)

Line 449: Please say what statistical value is that e.g. correlation coefficients?

**We will add this during revisions. You are correct that these values are correlation coefficients calculated using Pearson's r.**

- Added clarifying language (L919).

Line 489: The reason why this study found an overestimation is not that more grains have been analyzed, but the reported measurements are based on the maximum width and assuming an idealized grain geometry. You state in the method section that you also did minimum width measurements. In case you have the data, I would like to see the resulting measurements for those too, or at its best to combine both maximum and minimum measurements.

**We will reword the sentence to avoid implying that we found an overestimation due to the number of grains measured. As noted above, we also will add the data, figure, and discussion regarding the use of one or both widths.**

- We reworded this section (L1085-1089) to clarify our point. As noted above, we included the minimum width results in Table 3, Appendix D, and the raw data can be found in our data repository on OSF.

Line 532-533: This is because I did the calculation based on the minimum and maximum width of grains or accurate measurements of the a, b and c axis.

**Thank you for this important point. We will clarify that you measured all three dimensions and used these in your calculations when comparing our results to these outcomes.**

- Clarified this point on L1112 and L1145.

Figures

Fig. 1: Please explain the occurrence of the overestimated area. Is this based on the assumption that W1 is measured, but why in the ellipsoid example do you fit an asymmetric cross section (taking into account that W1 and L are different)? Please clarify.

**Thank you for your suggestions. We will rework Figure 1 based on your and Dr. Ketcham's reviews.**

- Redid Figure 1 as described above (L130).

Fig. 4: Could you use a slightly different color for those grains in morphological categories 1 or 2.

**We are unsure which figure you are referring to since Figure 4 is a black and white photomicrograph, but would be happy to modify the color scheme on the figure you meant.**

- All figures with photos are in black and white.

---

## Author Response (AR2)

Dear Editor,

Thank you again for your prompt and careful decision on our manuscript. For your convivence, here is a list of relevant changes made in the manuscript:

Text changes:

- Throughout text changed "ellipsoid" to "ellipsoidal" (per Rich Ketcham's comment)
- Throughout the text (after Sect. 6.3) changed the subscripts to reflect the change from the "orig" and "new" subscripts to "2D" and "GCM" (per Rich Ketcham's comment)
- L18: removed hyphen
- L106: changed "renders" to "renderings from CT data"
- L149: changed "mineral diffusion" to "helium retention" and edited sentence to say "…helium retention can depend on grain size, which must therefore be…"
- L160: changed software citation style (per Rich Ketcham's comment)
- L220: changed "is based on" to "was designed to be representative of…"
- L313-322: added a sentence and clarified previous text to describe how we measure the minimum width of apatite grains (per Christoph Glotzbach's comment)
- L399: changed software citation style (per Rich Ketcham's comment)
- L414: changed "Like" to "As"
- L431: changed "like" to "such as"
- L503-508: added a sentence to describe why our 2D minimum width measurements are smaller than the 3D minimum width measurements (per Christoph Glotzbach's comment)
- L514: added text to figure caption to clarify 1:1 line.
- L544-546: added a sentence to address how our measurement of the minimum width impacts the "both widths" correction.
- L1094: made "grains" singular (grain)
- L1105: added a bullet point under Step 2 to clarify why measuring your grain is a necessary step.
- L1591: updated He and Reiners (2022) citation to include journal information.

Figure changes:

- Figure 1: changed "ellipsoid" to "ellipsoidal" in the labels
- Figure 3: changed "ellipsoid" to "ellipsoidal" on the axes
- Figure 4: changed "ellipsoid" to "ellipsoidal" in the labels
- Figure 6: Updated the axes so that for each plot the x and y axes are the same, making it clearer that the line is a 1:1 line & changed "ellipsoid" to "ellipsoidal" in the legends
- Figure 7: changed "ellipsoid" to "ellipsoidal" in the legends
- Figure 8: Updated the axis titles to reflect the change from the "orig" and "new" subscripts to "2D" and "GCM".
- Figure 9: Made clarifying changes (removed the split in Box 2; added text to Box 4 and 5) based on Rich Ketcham's comment
- Figure B1: changed "ellipsoid" to "ellipsoidal" on the axes

- Figure C1: changed "ellipsoid" to "ellipsoidal" in the legends
- Figure D1: changed "ellipsoid" to "ellipsoidal" in the legends

Table changes:

- Table 3: changed "ellipsoid" to "ellipsoidal"
- Table 4: changed table headers, table title, and table caption to reflect the change from the "orig" and "new" subscripts to "2D" and "GCM".
- Table E1 (L1338-1449): changed table headers, table title, and table caption to reflect the change from the "orig" and "new" subscripts to "2D" and "GCM".

Below you will find our point-by-point response to each of the reviewer's comments in bold.

Thank you,

Spencer Zeigler

Report #1: Christoph Glotzbach

**Dear Dr. Glotzbach,**

**We would like to thank you again for your prompt, detailed, and thorough review. Below, we respond to your points with the line numbers of the changes.**

Dear Authors,
First of all, I would like to thank you for seriously going through my comments and corrections (and those of the other reviewer). I only have a single point that I would like to be addressed by the authors:
Thanks for showing your minimum width measurements compared to the minimum values from the CT data, but why are your manual measurements smaller than the CT data? 2D microscopic measurements cannot be smaller. I agree with you that it is difficult to orient the crystals to see/measure the minimum width and that it is very likely to measure something in-between the maximum and minimum width (but not smaller!). Since your measurements of the minimum width are 'too small', the resulting V, FT and RFT underestimate the real values. You are using this to argue in the discussion that it does not reduce the inaccuracy of the data, but instead, corrections and uncertainties are higher. Please explain why you have 'too small' minimum widths, which, as this point, shows me that you did something wrong during the measurement. Please clarify and correct if necessary, and based on this, I would ask you to adopt your argumentation.
**We are glad you brought this point to our attention! In order to address this comment, we added clarifying language to several sections:**

**Sect 4.2 (L313-321)**
**"This was followed by attempting to roll the grain 90°, acquiring another grain photograph, again measuring the long axis using the Leica software to obtain a second**

length measurement, and estimating and measuring the apatite's "minimum width". Typically, the minimum width that was measured was less than the observed width following grain rotation, because it challenging to efficiently position the grain such that its true minimum width is perfectly visible for measurement in the field of view. This difficulty of rotating and stabilizing the grain for a photograph while the grain is balanced on its minimum width axis makes it difficult to determine and measure the apatite's minimum width accurately. For rounded grains (GEM C, ellipsoidal idealized geometry), the length and widths can be particularly difficult to identify."

**Sect 5.1 (L503-509)**

"The systematic 2D underestimation of the minimum width is because the analyst was aware that the observed width of the grain after attempting to roll it 90° from the maximum width position (Sect. 4.2) was larger than the apatite's actual minimum width, and then overcompensated for this fact by acquiring a measurement that was not only smaller than the observed width but also inadvertently smaller than the true minimum width."

**Sect 5.2 (L544-546)**
"The underestimation of 2D values when using both widths is due to microscopy measurements that systematically underestimate minimum width values (see Sect. 5.1)."

I hope you find my comments and suggestions helpful.
**We did, thank you!**

Technical corrections:

Line 462-464: You stated before that it is difficult to measure the minimum width of a grain, and I fully understand that. But why are your 2D measurements smaller than the CT data's minimum width? It should be the opposite, and I hope you have a reason for it (I do not have one). This also leads to underestimating the volume, Ft and SER in Fig. D1.
**See above response.**

Line 494-496: The reason for this is very likely the 'too' short measurements of the minimum width (see the comment before).
**Thank you for this point; we agree and have added a sentence to section 5.2 (L544-546) to clarify this, as noted above.**

Report #2: Rich Ketcham

**Dear Dr. Ketcham,**
**Thank you for your prompt and helpful review.**

That additions have improved the paper, and I recommend it's publication. I only have a few final, minor comments and suggestions.
Line numbers are based on the manuscript with changes marked (egusphere-2022-1005-ATC2)

[line 19] Replace hyphen in "computed-tomography" with a space.
**Done.**
[line 129] Change "renders" to "renderings from CT data"
**Done.**
[line 190] Change "mineral diffusion" to "helium retention"
**Done.**
[line 191] Consider changing to "… can depend on grain size, which must therefore be included…"
**Done.**
[line 202] Software citation still odd. I recommend giving the citation for each after the name on line 201, and using (Object Research Systems, v.2020.2) for Dragonfly.
**Thank you for catching this again, we have made this change!**
[line 283] Consider replacing "…is based on the size distribution…" with "was designed to be representative"
**Done.**
[line 515-516] Same comment as for line 202
**Done.**
[line 530] Change "Like" to "As"
**Done.**
[line 544] Change "like" to "such as"
**Done.**
[Figure 6 caption] What are the lines – 1:1, or fits? Please specify (it's hard to tell because the x and y axes don't match).
**Done.**
[line 701] Here you use "ellipsoidal" rather then "ellipsoid", as in the rest of the paper. You can either change this one, or change all the others. "Ellipsoidal" seems preferable, as it's an adjective.
**We agree with your suggestion and the change to 'ellipsoidal' has been made throughout the text, figures, and tables.**
[line 1012] "New" is sort of a funny term and subscript to use in the subsequent text and figures, as it's not very descriptive; it's OK in the immediate context of this paper, but it would odd down the road to report one's measurements as "new". Maybe something to indicate it's a corrected value would be better, such as a "corr" subscript, or even "GCM".
**Excellent suggestion! We have made this change throughout the text/figures.**
[line 1175] Change "grains" to "grain", as you use the singular for the rest of the description.
**Done.**
[Figure 9] Why does the Step 2 box separate medium and large grains? I can't see where that distinction changes how the grain is subsequently treated; if it doesn't, it's confusing to have it in the workflow. If it does change something, specify it.
**We agree that this is confusing, we made clarifying changes to both the figure and the accompanying text.**